# Similarities and differences in waste composition over time and space determined by multivariate distance analyses

**David J. Tonjes**[ORCID]\*, **Yiyi Wang, Firman Firmansyah, Sameena Manzur, Matthew Johnston**[¤]**, Griffin Walker, Krista L. Thyberg**[ORCID]**, Elizabeth Hewitt**

Waste Data and Analysis Center, Department of Technology & Society, Stony Brook University, Stony Brook, New York, United States of America

¤ Current address: Middlesex County Utilities Authority, Sayreville, New Jersey, United States of America
\* david.tonjes@stonybrook.edu

## Abstract

The composition of solid waste affects technology choices and policy decisions regarding its management. Analyses of waste composition studies are almost always made on a parameter by parameter basis. Multivariate distance techniques can create wholisitic determinations of similarities and differences and were applied here to enhance a series of waste composition comparisons. A set of New York City residential waste composition studies conducted in 1990, 2004, 2013, and 2017 were compared to EPA data and to 88 studies conducted in other US jurisdictions from 1987–2021. The total residential waste stream and the disposed wastes in NYC were found to be similar in nature, and very different from the composition of wastes set out for recycling. Disposed wastes were more similar across the five NYC boroughs in a single year than in one borough over the 28-year time period, but recyclables were more similar across 14 years than across the boroughs in a single year. Food and plastics percentages in total and disposed waste streams increased over time, and paper percentages fell. The food disposal rate in NYC over time increased much less than EPA data show. The rate of plastics and paper disposal in NYC decreased. NYC data largely conformed to trends from the 88 other waste composition studies and did not generally agree with EPA data sets. The use of novel-to-waste studies multivariate distance analyses offers the promise of simplifying the identification of overall similarities and differences across waste studies, and so improving management and planning for solid waste.

## Introduction

The solid residues produced at residences, businesses, and institutions are called municipal solid waste (MSW) [1]. Current annual worldwide MSW generation is approximately 3 billion tonnes, which is a per capita generation rate of 1 kg/d, based on a population of 7.5 billion [2, 3]. Managing these wastes causes environmental problems, including methane as a greenhouse gas released from landfills due to anaerobic decay of MSW [4], plastics as ocean

**Data Availability Statement:** Data sets extracted from the waste composition studies and the analyses of the data sets are posted at: https://wastedata.info/data-repository.

**Funding:** DJT, YW, FF, SM, MJ, GW, KLT, and EH were funded by New York State Environmental Protection Fund as administered by the New York State Department of Environmental Conservation through a Memorandum of Understanding with Stony Brook University (AM 11643) effective September 11, 2019. The sponsor played no role in the study design, data collection and analysis, decision to publish or preparation of the manuscript.

**Competing interests:** The authors have declared that no competing interests exist.

pollutants [5, 6], and nano/microplastics incorporated in the biosphere [7]. Effective waste management is constrained in many ways by the composition of MSW. The value of collected materials for informal recycling is determined by the metals and plastics content of the wastes [8]; the percentage and energy value of combustible components of the waste stream identifies systems where waste-to-energy incineration is a better option [9]; the organic, readily degradable content of landfilled MSW controls the generation of landfill gas, both as a potential pollutant and as an energy source [10]; and the composition of wastes affects the potential for development of more esoteric energy recovery processes, including pyrolysis, anaerobic digestion, and gasification [11]. Generally, developing countries tend to have waste streams with higher proportions of organic materials, and management is mainly by landfilling, with less developed formal recycling programs. In more developed countries, where there may be more fabricated materials in the waste stream, there tend to be better and more widely organized recycling programs, and incineration with energy recovery may be used as a means of managing wastes. Often governments will specify the management of particular waste streams (such as yard wastes, food wastes, or recyclables) [3]. Even in countries with strong environmental regulatory programs, the composition of MSW is not necessarily well understood and is often poorly described in only the most general terms.

In the US, the most common reference for MSW management, generation, and composition is a series of reports published annually by the US Environmental Protection Agency (EPA) (e.g., [1]). These reports are based on a model created and maintained by a consulting firm (Franklin Associates) that blends an economic input-output model with other elements derived from industry reporting and field measurements. EPA explicitly states that the model is meant to represent the US waste stream as a whole and is unlikely to be representative of MSW at any particular location [12]. It has been found that the size of the US waste stream described by the model does not match compilations based on state and local recordkeeping [13–15] and its overall modeling approach has been said to be theoretically flawed [16]. Some parameters of the model have been measured by others using independent approaches, notably food waste [17, 18], plastics [18], and paper [19] where the estimates of disposed amounts were found to be substantially different from EPA's values. In contrast, a recent broad evaluation of EPA data sets compared to waste composition data from field studies generally found good agreement except for paper [20]. Overall, the EPA model remains the primary source for waste composition estimates for the entire US waste stream and its data are used to power US government sponsored models that estimate methane generation in landfills and for environmental impacts from waste management activities (for instance, as used in [21]). Thus, the EPA data sets influence climate impact policies on local, national, and international scales, despite some claims that the size and composition of the EPA waste stream are not accurate (e.g., [16]).

Various states, cities, and other municipalities have sponsored waste stream composition studies, generally in support of planning efforts, since the late 1980s. Some of these studies simply apply EPA composition data to locally measured waste quantities (e.g., [22]). Some combine results from waste sorts conducted elsewhere but thought to be representative of local conditions [23, 24]. Florida created a modeling approach grounded in certain local waste sorts where projections for the years after the original studies are generated based on adjustments made for local economic and demographic information and annual waste generation and recycling data [25]. Others use generator estimation functions for types of waste streams to create composition and amounts of amalgamated streams (e.g., [26, 27]). Recently, neural networks have been used to analyze socio-economic data and their relationship to waste types and overall waste generation rates to generate models of current and future waste composition [28, 29].

However, most waste composition studies physically sort samples of MSW into predefined categories to determine waste stream composition. Although guidance to conduct such

sampling has been published [30], the exact procedures set out by the American Society for Testing and Materials are generally followed only in the broadest sense by the organizations that do this kind of work. Nonetheless, these studies have a somewhat consistent approach. A set of replicate samples intended to represent some aspect of a waste stream of interest is selected; each sample is approximately 100 kg (200 lbs) and has been separated from the larger mass of wastes by a process usually said to be random but in practice mostly best described as "haphazard" or "undirected." Each sample is hand sorted by piece by trained teams into categories. The number and exact nature of the categories vary across studies. Publications report MSW composition as percentages of the whole, often with a measure of variability such as standard deviation or standard error that assumes that the data are normal. If the tonnage managed within the waste shed has been measured and reported, annual amounts for each category are estimated, and if an associated population of generators can be determined, the waste generation rate per capita or per household for waste stream constituents can be reported. That is, given a percentage value for a particular material, the amount of that material can be determined by multiplying the percentage by the total amount in the waste shed. Dividing the resultant amount of wastes by the relevant population and a time measure, such as 365 for a daily value, will create a rate datum. Thus, it is possible to collect a set of waste composition studies, determine if the procedures used to generate the data sets were congruent (the data are for disposal, or are all residential wastes, etc.), and then use the data sets in various combinations to extend the relevance of the studies. This has been done for specific waste types: food, using meta-analysis [17], and less rigorously for plastics [31] and paper [19]. Conceptually, other waste elements could be similarly examined. However, such single element studies do not create a depiction of the overall waste stream and do not allow quantitative determinations of how a particular waste stream description is different from or similar to other such descriptions, or how they compare in some wholistic sense to something like the "baseline" EPA data.

Staley and Barlaz [32] extracted 35 distinct waste elements to create a US waste composition using 11 state-wide waste composition studies that all measured disposed waste; however, not all of the included studies had all 35 waste stream components. The compiled mean data were qualitatively compared to EPA data for 2006 and were used to estimate the methane generation potential of disposed wastes. Kumar and Garg [20] recently used 27 studies from 24 states and three smaller jurisdictions to represent waste characterization percentages from 1997–2019. Population weighted averages were used to compare to EPA data when more than one study occurred in the same year, and the yearly estimates were averaged for 2010–2017 under an assumption that national waste disposal was flat over that time period. The overall conclusion was that approximately 15 million more tons (13 million tonnes) of paper is disposed each year compared to EPA estimates, although the other materials estimates were not as different.

Here we will examine four waste composition sort studies for a particular jurisdiction (New York City) conducted in 1990, 2004, 2013 and 2017. Because these studies of residential wastes not only sorted wastes collected for disposal but also recyclables it was possible to also depict the "generated" residential waste stream, which is not common in these kinds of studies. The percentage and rate data and associated trends for three broad materials categories (paper, food, and plastics) will be compared to EPA data and trends for the same years; the comparisons for disposal will be put into context using 88 other waste composition sort studies from 1988 to 2022. The sort study data trends generally show differences from the EPA data trends.

In addition, we will pioneer a more wholisitic waste stream composition approach that uses the basic techniques that drive machine learning. We will look at the multivariate distance relations among the result sets and use the principle that points close to each other in multivariate space are more similar and points that are farther away are different. This will allow us to

underscore several common concepts about overall waste composition for different manage-
ment processes (disposal and recycling) and to tentatively identify testable theses regarding
changes in waste composition over time and space. These results will confirm that EPA mod-
elled composition data tend to be different from composition results generated by physically
sorting MSW.

## Materials and methods

### Waste composition studies used

We collected four waste composition studies conducted in NYC in 1990, 2004, 2013, and 2017.
NYC is the largest city in the US. Its budget for residential waste management (and snow clear-
ance) in 2021 exceeded $1 billion; its landfill received 9 million tonnes per year of MSW circa
1988 and the city manages approximately 12,000 tonnes per day as of 2020, consisting of resi-
dential and some institutional disposed wastes and recyclables. The four studies were con-
ducted over a 28-year period by different organizations; officials at NYC Department of
Sanitation (DSNY) have generally represented the studies as being "not comparable" because
specific categories and groupings of data are not consonant. Nonetheless, we have constructed
seven broad categories that are common across the studies: paper, glass, plastic, metals, food,
yard waste, and other (that is, the balance). The categories include all subcategories (if any) of
the waste sorts. So "plastics" here includes all containers of various resin types, plastic films,
and all durable plastics. The categorization created percentage composition of disposed, recy-
cled, and the total curbside collection waste streams. We used the individual boroughs as well
as data for NYC as a whole but did not include available 1990 and 2004 seasonal data sets. The
seasonal data sets were only available for half of the studies and generated data depiction com-
plications, such as how to show annual values in the context of the four seasonal data points.
The multiple closely packed data points across nine months in 1990–1991 and 2004–2005
made trend determination unduly complicated. Using tonnage data and population size esti-
mates we were able to construct annual rates for the NYC data sets. The analysis also includes
comparable EPA waste composition data for the four years. Details on data use from the NYC
and EPA studies are in S1 File.

The secondary analysis compared reported NYC disposal data for 1990-2004-2013-2017
and EPA disposal data with a wider array of waste composition sort studies reporting disposal
data. The EPA data sets used were all studies from 1988–2018 for the individual parameters,
and data for 1990-2004-2013-2017 for the multivariate analyses. The comparison studies were
selected in three ways. One was a set of nine studies chosen by RW Beck for the 2004 NYC
study to compare to the 2004 NYC data sets; the second was 10 waste composition studies
from other New York State jurisdictions (Albany NY, Town of Brookhaven NY, Monroe
County NY, and Onondaga County NY); and, finally, we selected 60 reports from a set of 215
waste composition sort studies dating from 1987 to 2022. The criteria for selection was that
each jurisdiction had to have at least four replicate studies. This resulted in data sets from Cali-
fornia, Clark County WA, Iowa, King County WA, Missouri, Orange County NC, Oregon,
Pierce County WA, Seattle WA, Thurston County WA, and Washington State. This gave us a
total of 88 comparative studies. See S2 File for references to the studies and S3 File for more
details on data used from these 88 studies. S4 File lists all of the EPA studies used to generate
the 1990–2017 data sets.

### Standard analyses

Every solid waste composition study that we have examined compares its results, either inter-
nally across categories or to other years for the same location, or to other studies, by single

variables, with two exceptions [33, 34] where non-parametric multivariate analyses were made using PERMANOVA [35] to determine if significant differences existed across the entire waste stream. The first study determined that there were significant waste stream composition differences across three different waste districts in a Long Island municipality [33]. The second found that waste composition had not significantly changed for those districts two years later although significant differences across districts were still found, in a broader examination of the impacts of changing from dual stream to single stream recycling [34]. In all other cases, simple parameter to parameter comparisons were made. These kinds of analyses are powerful in that they determine the relative or absolute prevalence of materials in the waste stream.

We have made these kinds of individual parameter comparisons and report here on food, plastics, and paper results. We have used EPA data points as comparisons to the NYC data. Additionally, we have the other 88 sets of samples described above to compare NYC mean data to other jurisdictions. For visual comparison we used all 30 EPA reports from 1988–2018 for disposed paper, plastics, and food. When we computed linear regressions we used the data from the NYC means and the 88 comparison studies. Regressions were computed using the Excel function.

## Distance measures

We used certain machine learning approaches and multivariate data displays and analyses to look at the results more wholistically. "Machine learning" encompasses a loose set of techniques that uses algorithms to find similarities and differences in data, usually large sets of multivariate data. Data are defined as similar when they have smaller differences and dissimilar when the differences are larger. Defining the data in a cohesive numerical framework and then measuring distances between elements of the data sets is the most common means of defining the difference relations.

Often the relations are visualized using techniques such as heat maps where color is used to identify similarities and differences in a broad qualitative fashion. In addition, statistical analyses such as multidimensional scaling (MDS), principal component analysis (PCA), and various forms of cluster analysis lend themselves to simplified visual depictions of similarities and differences across multivariate data sets such as the waste composition studies. Reducing the dimensionality of multidimensional data sets each have relative advantages and issues. Multivariate correlational analyses can also find similarities and differences across these kinds of data sets [36].

It is axiomatic that two mappings are more similar if they map closer to each other than if they map farther away. Thus, machine learning systems using neural networks to self-program identities determine a set of descriptors that distinguish the class. For visual identification of cats, for instance, this may involve the number of ears, the size of the ears, the distance of one ear from another, a roughness measure intended to define fur, etc. Each measure can be different in kind from one another, and so require parameter weightings or parameter translations to make measurements commensurate or to assign emphasis to aspects of the measures. Here the measurements are of the same general kind: percentages, bounded from 0% to 100%; or rates, which are theoretically unbounded but actually range between 0 kg/p/yr to 1,000 kg/p/yr. Therefore, our determinations of close or far away did not require axial weightings or other transformations of the base data sets.

**Euclidean distances.** Euclidean distance is the shortest distance between two points. An alternative measure is "Manhattan" distance, which is the sum of the absolute value of each axial measure. Each axis is orthogonal, so this measure is a multiple dimensional analogy to moving along right-angled blocks in a gridded city, akin to the Manhattan street grid, giving it

its name. Thus, Manhattan distance is the sum of the differences between each parameter for our data sets [37], a series of parameter-by-parameter comparisons and so of a kind to standard solid waste composition explanations. Euclidean and Manhattan distances for the same data sets are not the same nor are they isomorphic (see S5 File Comparisons of Euclidean and Manhattan distances for the NYC data). Here, we use Euclidean distance measures; our choice was guided by our perception that Euclidean distance measures more directly measure differences for multidimensional space relationships. We saw there was a clear analogy between Manhattan distance measures and considering a series of standard, single component waste composition analyses. However, we thought that measuring the direct distance was a straightforward means of depicting multidimensional differences, and so Euclidean distances was our preference.

We computed the Euclidean distance pairwise between the 66 base NYC data sets (although the 1990 NYC total and disposed data are the same because there was no recycling then) and 12 EPA data sets for both percentage and rate (kg/p/yr) data. The third multivariate analysis used Euclidean distances between the four city-wide NYC disposal data sets, the comparable four EPA disposal data sets, and data from the 88 comparison studies. We computed certain relationships between elements of the first two analysis sets, and tested them for statistical differences using t-tests assuming unequal variances (using the Excel function). The third analysis set did not lend itself to general identification of groups of similar results and so our quantitative analysis was more limited in scope.

**Principal component analysis (PCA).**   Principal component analysis (PCA) is, essentially, repeated correlational analysis where the correlations are orthogonal to each other [36]. Therefore, any two correlations create the potential for a two-dimensional representation using two axes that are perpendicular to each other. Each successive correlation seeks to minimize the remaining variance. Therefore, the first two correlations create a two-dimensional representation which accounts for more variance than any two other correlations. This creates a dimensional reduction that preserves a determined amount of the differences found in the original $n$-dimensional space. A graph of these first two axes therefore enables a visual approximation of clustering in the original $n$-dimension spatial data, if sufficient variance has been accounted for. Correlational PCA analyses are most often used when the data are of different types or scales. All the data used here were "percentage" data ranging between 0% to 100% or kg/p/y data which are similar types and scales, so covariance analyses were conducted here. We used these covariance analyses to visually confirm similarities and differences found in the multivariate distance analyses of the seven parameter base NYC and EPA data sets for both percentage and rate measurements, and to further explore qualitatively relations among the city-wide NYC, EPA disposal, and comparison data set disposal data (in percentages).

## Results

### Standard analyses (single parameters)

The single parameter analyses cover food, plastics and paper. Food is a concern as a feedstock for landfill methane generation, among other attributes. Plastics create ocean and microplastic pollution and have more general mismanagement concerns. Paper is also a potential source of landfill methane, and, according to EPA data 1960–2018, the greatest single component of total MSW. NYC trends from the waste characterization studies for waste generation, disposal, and recycling will be summarized for each material, compared to corresponding EPA data sets, and then further compared to the disposal data from the 88 other waste characterization studies from 1987–2022, and EPA data sets over 1989–2018. Similarities and differences for the borough results will be briefly discussed.

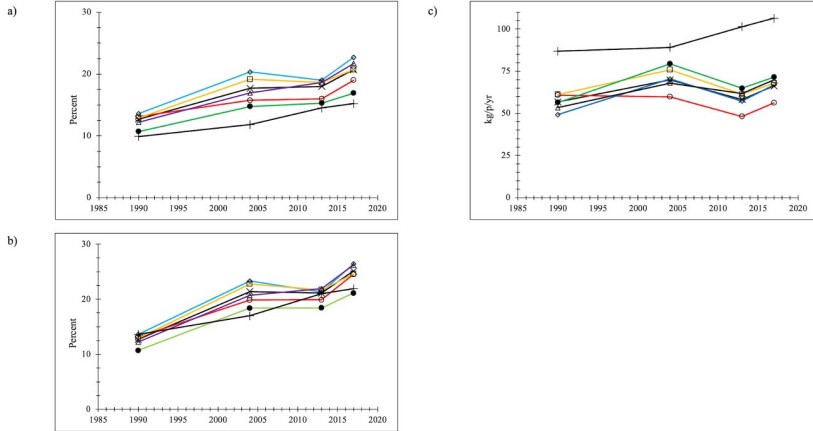

**Fig 1.** Food waste (a) percentage of the total waste stream; (b) percentage of the disposed waste stream (c) disposal rate (kg/p/yr) x = NYC (black line); o = Manhattan (red line); ◇ = Bronx (blue line); □ = Brooklyn (orange line); Δ = Queens (purple line); ● = Staten Island (green line); + = EPA (black line).

**Food.** New York City food waste percentages increased from 1990, both for the total waste stream and the disposed waste stream, even with flat trends or slight decreases from 2004 to 2013 (Fig 1a and 1b). Food disposal percentages are greater than the total waste stream food percentages because, for one reason, recycling (for the 2004-2013-2017 data sets) removes other materials from the waste stream but not food, thus making the same amount of food proportionately greater in the disposed samples. EPA data were consistently less than the NYC percentages. Food waste disposal percentages increased by 98% from 1990 to 2017 for NYC as a whole, greater than the 61% increase reported for EPA. NYC food waste disposal rates increased from 1990 to 2017 by 17% for the city as a whole (Fig 1c). From 2004 to 2017, however, there was a 6% decrease in food waste rates across NYC despite relative food waste percentage increases. This data anomaly of increasing food waste percentages and decreasing food waste rates is because overall total waste generation rates (kg/p/yr) declined in NYC from 1990 to 2017 by 27% and waste disposal rates declined by 41% for NYC as a whole, including a 19% decline from 2004 to 2017 [38] (see S6 File for the graphs). EPA food waste data showed increasing food waste rates although overall waste disposal per person decreased 7% from 2004–2017 [38] (S6 File) because the decreased disposal rate was outweighed by the increase in food waste percentages. NYC experienced a decline in disposed food waste per person from 2004–2017 despite not creating programmatic alternative disposal means for this waste stream constituent.

EPA changed its method of computing food waste for 2018 [39] to incorporate more empirical data and expand its universe of food waste generation and recovery pathways. EPA has determined that food waste rates are increasing, especially given its new methodologies for calculating food waste. Note that unlike other model alterations, the revised estimation methodologies were not applied to previous food waste generation years. All EPA reports carefully note that the data generated for the reports are national data and may not represent local waste patterns. Food waste data for NYC showed distinctly different trends than seen in the EPA data sets, at least partially because waste generation and waste disposal rates were decreasing more in NYC than in the national EPA data.

Food waste showed an increasing trend over time across the comparison data sets (Fig 2a). The slope of the regression line was significantly positive (p < 0.001). There was a great deal of scatter in the data. The $R^2$ coefficient is 0.13, suggesting the linear trend accounted for 13% of

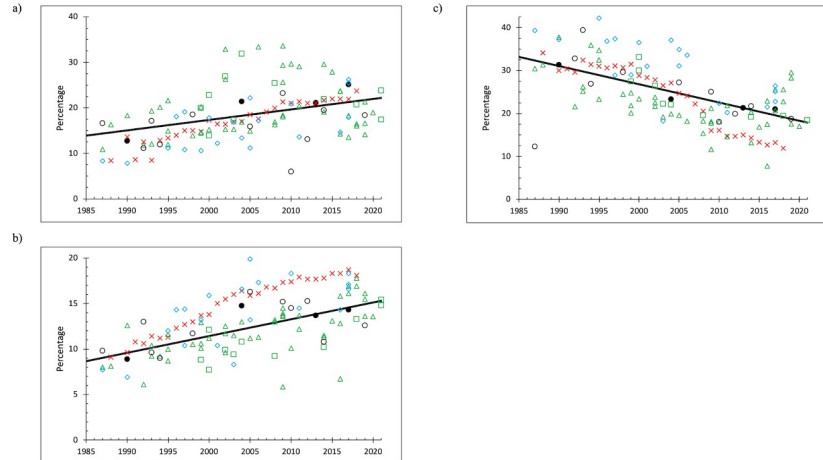

**Fig 2.** Disposal comparisons (a) food percentages (b) plastics percentages (c) paper percentages ● = NYC (in black);
x = EPA (in red); Δ = Washington-Oregon (in green); □ = California jurisdictions (in green); ◇ = other (in blue);
o = other New York jurisdictions (in black); solid black line = linear regression.

the data variability. The 1990 NYC percentage was below the regression line and the three
other NYC data points were above the line, so that the increase in food waste as a percentage
of the disposed waste stream was greater in NYC than for the rest of the country. EPA data
were below the regression line until the 2000s, and then were slightly above the regression line.
The input-output economic model cannot address food waste, so it is clear that EPA must use
information sources similar to waste composition reports to generate its food waste data.
Therefore, conformance with a broad array of waste composition studies might be expected.
Many of the "West Coast" data points in green triangles and squares in Fig 2a (Washington
plus jurisdictions, Oregon, and California plus jurisdictions) were considerably above the
regression line, and most of the "other" non-New York data points (Orange County NC, Phila-
delphia PA, Pennsylvania, Phoenix AZ, Georgia, Missouri, and Iowa, hereafter "other") were
below the regression line. A number of the West Coast data points were also below the regres-
sion line albeit lying approximately on the linear trend. The data show that the NYC data sets
were similar in pattern to waste composition data from other jurisdictions and imply that the
EPA data set conforms to food waste percentage trends for the country as a whole.

For boroughs, food waste generation percentage increases from 1990–2017 ranged from
45% in Manhattan to 94% in the Bronx (+94%), with flatter rates from 2004 to 2013. Waste
generation rates (kg/p/yr) declines ranged from 17% in the Bronx to 36% in Manhattan. The
changes across the boroughs, as illustrated by the many parallel line segments in Fig 1, were
relatively similar.

**Plastics.** Plastics production and use have been increasing around the world since the
1960s and the NYC data sets agree with these broad trends. The percentage of plastics in the
NYC total waste stream increased by 67% since 1990 (Fig 3a), from 8.9% of the total waste
stream to 14.9%. EPA data were similar: a 57% increase since 1990 from 8.4% of the generated
waste stream to 13.2%. The EPA increases were nearly linear but for NYC growth in plastics as
a percentage of the waste stream stalled beginning in 2004, increasing by 3% from 2004 to 2017
(14.5% to 14.9%). The plastics waste generation rate (kg/p/yr) in NYC increased from 1990 to
2017 by 22% (Fig 3c). "Peak" plastics waste generation for NYC as a whole was 2004 (by rate,
in kg/p/yr), decreasing by 13% by 2017. From 1990 to 2017 EPA data had a 58% increase,
although growing only 8% from 2004 to 2017. EPA plastics amounts were approximately

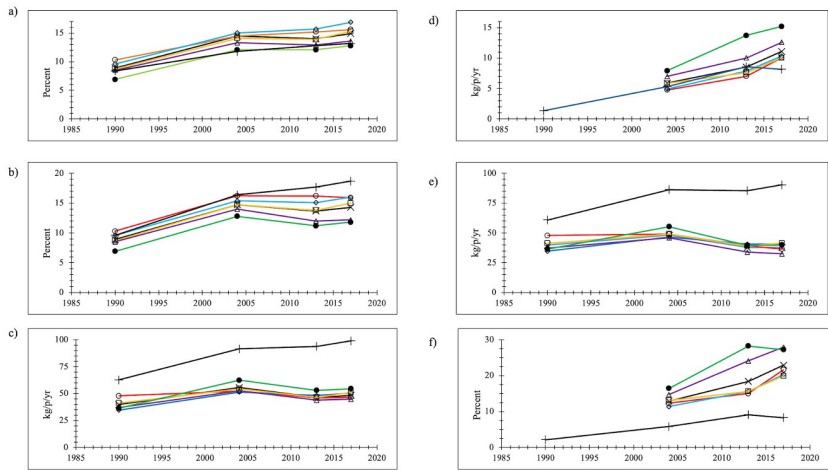

**Fig 3.** Plastics (a) percentage of the total waste stream; (b) percentage of the disposed waste stream (c) waste generation rate (kg/p/yr) (d) recycling rate (kg/p/yr) (e) disposal rate (kg/p/yr) (f) recovery percentage x = NYC (black line); o = Manhattan (red line); ◊ = Bronx (blue line); □ = Brooklyn (orange line); Δ = Queens (purple line); ● = Staten Island (green line); + = EPA (black line).

double NYC amounts, perhaps partially because EPA included commercial and institutional data in its waste stream assessments whereas the NYC data were only residential data. For NYC, although plastics were increasing as a percentage of the total waste stream, the waste generation rate (kg/p/yr) for plastics from 2004–2017 declined.

Curbside recyclables collection became available in all parts of NYC by mid-1992 (there was a progressive program expansion beginning in mid-1990). The first data for plastics recycling impacts on waste composition were from the 2004 waste composition study. Early recyclables collections were dominated by paper but plastics recycling grew over time. Plastics recovery grew from 12.9% in 2004 to 22.9% (a 77% increase) (Fig 3f) and the per capita recovery rate increased from 5.9 kg/p/yr to 11.1 kg/p/yr (Fig 3d). EPA plastics recovery also grew from 2004, but the percentage of plastics recovered was much lower than the NYC data, and the rate of recovered plastics (in kg/p/yr) fell slightly from 2013 to 2017, becoming less than the NYC rates.

Plastics as a proportion of the disposed NYC waste stream increased by 61% overall by 2017, from 8.9% of the disposed waste stream in 1990 to 14.3% in 2017 (Fig 3b). EPA data showed the proportion of plastics in waste disposal doubled since 1990 (from 9.6% to 18.7%). Plastics disposal rates for NYC in 1990 were 39.8 kg/p/yr compared to 60.9 kg/p/yr for EPA (Fig 3e). NYC plastics disposal rates peaked in 2004 at 48.1 kg/p/yr, then declined so much so that each resident was discarding slightly less in 2017 (37.8 kg/p/yr) than they had in 1990, 22% less than 2004. EPA data continuously increased to 90.5 kg/p/yr, a 49% greater disposal rate. The rate of change slowed so that the increase in disposed plastics from 2004 to 2017 was only 5%. The difference in the NYC disposal data from 2004 to 2017 (10.4 kg/p/yr) was not entirely due to increased recycling (+5.2 kg/p/yr 2004–2017). People in NYC were using less plastics overall. A similar trend was not found in EPA data.

For the comparison disposal data sets (Fig 2b) the regression trend was significantly positive (p<<0.001) showing increasing plastics disposal percentages over time. The $R^2$ coefficient was 0.33 and visual inspection shows less scatter in the data compared to the food data in Fig 2a. Three of the NYC data points (1990, 2013, and 2017) fell almost exactly on the linear trend, and most of the New York state data points were also found close to the trend line. The West

Coast data tended to be below the trend line and the other data tended to be found above the trend line. EPA disposed plastics percentages were greater than the trend line (except in 1988 and 1990), and the difference grew over time. The percentage disposal data for plastics for NYC therefore accord with "national" trends, but EPA data trends do not appear to match the waste composition data.

Plastics comprised a slightly greater proportion of Manhattan and the Bronx waste generation and disposal percentages, and less of Staten Island's wastes. However, because Staten Island has a greater total waste stream size per person than other boroughs, more plastics were generated, disposed, and recycled in Staten Island than in other boroughs. Staten Island and Queens had higher recovery rates and recovery percentages than the other boroughs, which resulted in Queens disposing of less plastics than the other boroughs. All of the boroughs exceeded 20% recovery of plastics by 2017 except Brooklyn (19.9%), much higher than the EPA 2017 8.3% rate, and the oft-quoted datum of 6% plastics recycling. The NYC data reflect sampling data that reported what was in the disposed waste stream and what was collected as recyclables. These data were then multiplied by the amounts of each stream, and then compared to each other to determine the ratio of recycled materials to the total amount in the waste stream. EPA uses its own methods to collect recycling amounts nationwide, generally assumed to be from industrial and waste recovery trade organizations. It then divides the material waste generation amounts by the recovery amounts to generate its recovery percentages.

**Paper.** In 1990 paper was the largest element of the NYC waste stream at 31.3% and in 2017, although 15% less at 26.6% of the waste stream, it was still greater than food (Fig 4a). For EPA paper decreased from 35.5% to 25.0%, a 30% decrease. The decrease in the rate of paper waste generation (in kg/p/yr) for the NYC total waste stream was 38.8% from 140 kg/p/yr to 86.4 kg/p/yr (Fig 4c). The decrease for EPA was somewhat less (30%). Part of the decrease in paper in the waste stream would appear to be that newspaper readership plummeted, much more so than increases in on-line shopping which generate additional corrugated cardboard paper boxes.

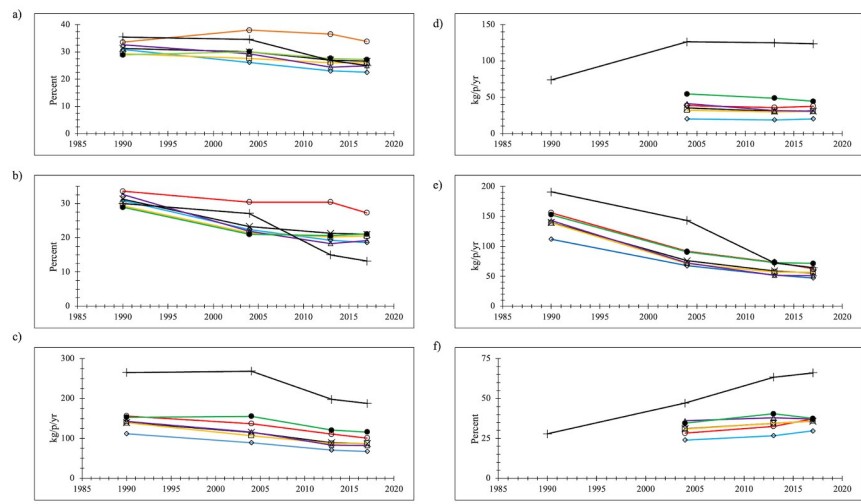

**Fig 4.** Paper (a) percentage of the total waste stream; (b) percentage of the disposed waste stream (c) waste generation rate (kg/p/yr) (d) recycling rate (kg/p/yr) (e) disposal rate (kg/p/yr) (f) recovery percentage x = NYC (black line); o = Manhattan (red line); ◇ = Bronx (blue line); □ = Brooklyn (orange line); Δ = Queens (purple line); ● = Staten Island (green line); + = EPA (black line).

Paper recycling increased from nothing in 1990 when there was no NYC curbside recycling program to 31.0 kg/p/yr in 2017 (Fig 4d). Although the rate per person of paper recyclable collection was greater in 2004 than in 2017 (35.2 kg/p/yr), the recovery percentage for paper was greater in 2017 than in 2004 by 15%, from 31.1% to 35.8% (Fig 4f). Paper was by far the dominant material in recyclables setouts in 2004, but by 2017 paper and MGP setouts were approximately equivalent. In 2017 the paper recycling rate per capita was more than three times greater in EPA data than calculated for NYC, although the amount of paper recycling for EPA was flat from 2004 to 2017. Nonetheless, because waste generation of paper had decreased, the EPA recovery rate for paper rose from 1990 to 2017, reaching 66.0%, on the order of twice the recovery percentage calculated for NYC.

Decreases in the overall paper waste stream together with the introduction of paper recycling meant that the disposal of paper declined tremendously in NYC from 1990 to 2017. Paper was approximately one-third of discards in 1990 (31.3%) but 21.0% in 2017 (Fig 4b). For EPA the change was even greater, from 30.0% of the discards stream in 1990 to 13.2% in 2017. The amount of paper disposed in NYC decreased 60.0% from 140 kg/p/yr to 55.5 kg/p/yr (Fig 4e). The decline for EPA was much greater in absolute terms from 191 kg/p/yr to 64.4 kg/p/yr, a decrease of 125 kg/p/yr compared to the NYC decline of 85 kg/p/yr. The proportional change was only 10% greater than the NYC change, at 66.0% compared to 60.0%.

The comparison data set for disposal shows paper waste percentages decreased sharply over time (Fig 2c). The regression slope was negative and significant ($p \ll 0.001$) and the $R^2$ coefficient for paper was 32%, similar to the regression for plastics. The NYC data points were very close to the trend line, as were much of the other NYS data; much of the West Coast data were lower and much of the other data were greater. The EPA data points conform well to the trend line until about 2008 and then they are increasingly less than the overall trend.

For all boroughs, paper has remained the dominant material category in the total waste stream, except in the Bronx where paper was 22.5% of the waste stream in 2017 but food was 26.4%. Manhattan was a borough of paper waste: it had much higher paper waste generation percentages and disposal percentages. Staten Island had the most paper (measured in lbs/p/yr) and recycled more. The Bronx had the least paper and recycled considerably less than other boroughs. Paper recyclables trends were variable by borough: Queens (-26%) and Staten Island (-18%) saw sharp declines in recyclables setout amounts from 2004 to 2017 but for the other boroughs setouts were relatively constant (-1% to -4% declines). Because of declines in the overall waste stream size, even in Manhattan where the proportion of paper in the waste stream remained essentially constant, the amount of paper in the waste stream decreased by 36% (and the amount of disposed paper decreased by 60%). The proportion of paper in discards only decreased in Manhattan from 33.6% to 27.3%, a 19% decline, but in other boroughs declines were greater, ranging from 27% in Staten Island to 41% in Queens. The declines in disposal rates were fairly consistent across the boroughs, ranging from 53% in Staten Island to 64% in Queens. Therefore, although recycling decreased by a good amount in Queens and Staten Island from 2004 to 2017 (but less so in the other boroughs), disposal of paper across all of the boroughs declined tremendously comparing 1990 to 2017.

## Distance relations (multivariate considerations)

The multivariate data sets for the five boroughs and all NYC and EPA across total waste stream, disposed wastes, and recyclables created 3,003 unique Euclidean distance measurements. The 1990 NYC disposal and waste generation data were exactly the same and were counted twice as there were also no NYC recyclables data for 1990. The distances were calculated for both percentages and rates (kg/p/yr). Euclidean distances were also calculated for the

NYC and EPA and 88 disposal data sets used as the "comparison" data set (4,560 unique comparisons).

Determining how to present the data relationships was difficult; triangular matrices that display each distance datum are dense with thousands of numbers and hard to interpret. "Heat maps," colorized representations of data ranges, simplify interpretation. We used colors ranging from red to purple (in conformance with the visible light spectrum) to illustrate the distance differences, with red (shorter light wave lengths) representing shorter distances. For the first data set (the borough and all NYC and EPA data) the matrix was organized by the generic waste types (total waste stream, disposal, and recyclables) and then suborganized by boroughs, all NYC, and EPA, and then within these subgroups by year. The matrix for the disposal comparison data set was arranged in two ways. First, by geography: NYC data, New York state data, Washington state data, Oregon, California state data, Orange County NC, the spare single results, Iowa, Missouri, and then EPA (and subsorted alphabetically and then by date). Secondly, by date of study: then subsorted using the geographical ordering listed just above. Qualitative analysis of the heat maps was used identify data relations that might yield interesting quantitative relations. Two-dimensional PCA representations were used to confirm the multivariate distance relationships extracted from the qualitative and quantitative analyses.

**Borough-all NYC-EPA multivariate distance measures.** The pairwise distance relations were visualized through heat maps (Figs 5 and 6). The two graphs share many features. Each has a large triangle of red-yellow occupying the entire left-hand space above the diagonal. These triangles in each figure are imprinted with regular square patterns. Continuing along the diagonal is another red-orange triangle. In Fig 5 this second triangle contains less yellow and no green points compared to the first triangle. In Fig 6 the second triangle has no yellow or green at all. The bottom right hand corner is a small red-yellow triangle. Above the bottom red-orange triangle is a solid color rectangle (green in Fig 5 and yellow in Fig 6). To the right

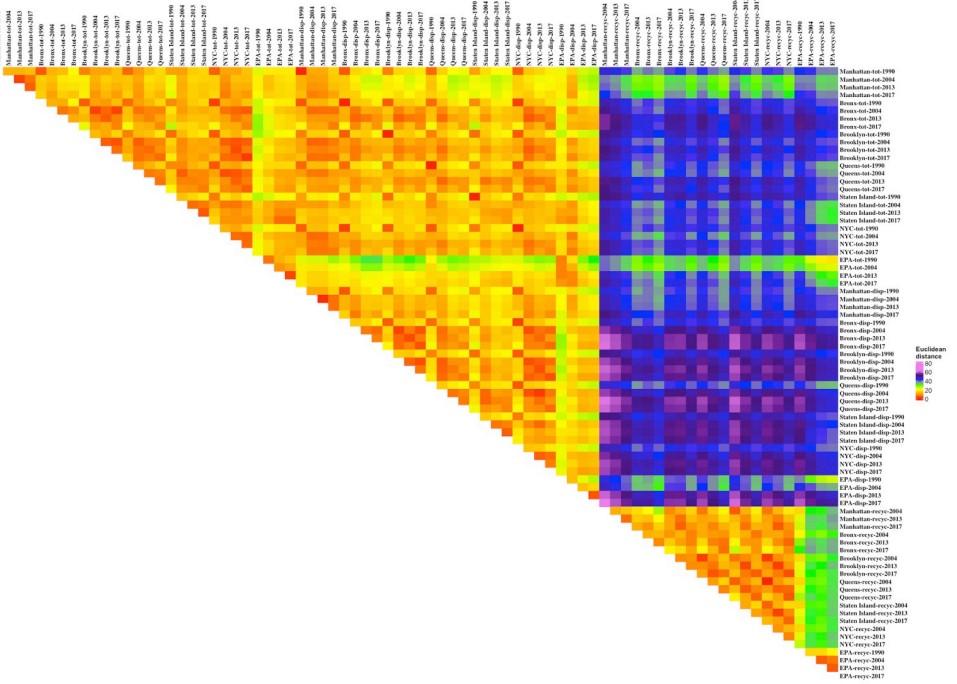

**Fig 5. Euclidean distances for percentages, boroughs and all NYC and EPA data.**

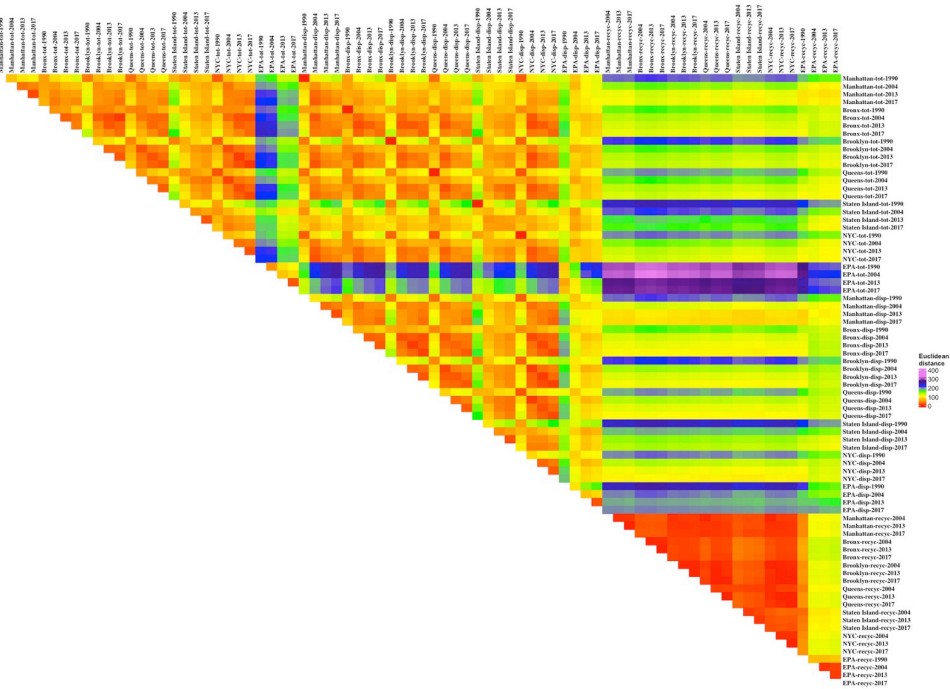

**Fig 6. Euclidean distances for rates (kg/p/yr), boroughs and all NYC and EPA data.**

of the large, first triangle is a large rectangle of similar colors in both Figures. In Fig 5 the rectangle is primarily blue and purple. In Fig 6 it is primarily yellow with green and blue elements and a band of purple.

These features are created by underlying data patterns. The large triangle is the distances among the waste generation data and disposed data groups and between these two groups. The red-orange colors show these data points are all close to each other, which suggests that not only are the different waste generation data points across the boroughs and for the NYC as a whole over the four sampling events similar to each other, but they are similar to waste disposal data from those events. The waste disposal composition data across the sites and over the four events are all relatively similar but also are similar to waste generation data. The squares in the data signify that particular borough data for waste generation are closer to each other than they are to other boroughs' waste generation data, and similarly one borough's disposal data across the four events are closer to each other than to the other boroughs' data. The spaced, repeated red squares also show that results in one borough for a particular sampling event (especially 1990) can be closer to other borough data. Certain Staten Island and some Manhattan data are marked by green points in the triangle in Fig 5. In Fig 6, some of the Staten Island results are marked by extended areas of yellow and green. This supports differences in waste composition for some of the Manhattan and Staten Island waste generation and disposal data.

The second triangle along the diagonal is the borough and NYC recyclables data compared to the other borough and NYC recyclables data. The lack of yellow and green squares indicates that the recyclables data are closer to each other than the waste generation and disposal data (especially for the rate data, Fig 6). The remaining, disjoint small triangle on the diagonal is the EPA recycling data compared across the sampling events. The data points are close to each other. Note that the EPA data in the waste generation and disposal triangle create the few

green points in that large triangle in Fig 5 and the EPA waste generation data in Fig 6 are marked by lines of blue across the triangle. EPA disposal data create a lighter yellow streak (with the EPA 1990 data in green) in Fig 6. This suggests that the EPA data for waste generation, disposal, and recycling are close to other EPA data in the same categories but not as close to the borough and NYC data in those categories.

The large blue-purple rectangle in Fig 5 is the comparison of waste generation and disposal data to recyclables data. The waste generation and disposal data map close to each other but far away from the recyclables data, suggesting that their waste compositions are distinct. The smaller yellow rectangle is the comparison of EPA recycling data to borough and NYC recycling data. They are not as close as EPA recycling data are to EPA recycling data or as borough and NYC recycling data are to other borough and NYC recycling data. This again supports a difference in waste composition across sampling events for EPA and borough-NYC data. In Fig 6 the relative differences between recyclables data and waste generation and disposal data are less, as marked by the overall yellow coloring, but EPA waste generation and disposal data are very different from the recyclables data as indicated by streaks of blue and purple. The 1990 data are also farther from recyclables data, as indicated by the spaced lines of blue.

This qualitative analysis can be supported by a more analytical approach (Table 1). Because 1990 total and disposed wastes were the same, the distance between them was always 0. All

**Table 1. NYC multivariate distance summaries.**

| Measure | Waste type | Comparison | N | Mean | SD | Min. | Max. | Different? |
|---------|-----------|-----------|-----|------|-----|------|------|-----------|
| **Percent** | **Total** | Within | 378 | 13 | 5.4 | 2 | 28 | |
| | | All others | 1400 | 26 | 14 | 0 | 56 | *** |
| | | Disposed | 784 | 15 | 5.8 | 0 | 33 | *** |
| | | Recyclables | 616 | 41 | 6.0 | 18 | 56 | *** |
| | **Disposed** | Within | 378 | 14 | 5.9 | 1 | 28 | |
| | | All others | 1400 | 29 | 18 | 0 | 63 | *** |
| | | Total | 784 | 15 | 5.8 | 0 | 33 | ** |
| | | Recyclables | 616 | 48 | 6.6 | 24 | 63 | *** |
| | **Recyclables** | Within | 231 | 17 | 10 | 1 | 36 | |
| | | All others | 1232 | 44 | 7.2 | 18 | 63 | *** |
| | | Total | 616 | 41 | 6.0 | 18 | 56 | *** |
| | | Disposed | 616 | 48 | 6.6 | 24 | 63 | *** |
| **Rates** | **Total** | Within | 378 | 90 | 59 | 8 | 241 | |
| | | All others | 1400 | 124 | 68 | 0 | 327 | *** |
| | | Disposed | 784 | 90 | 55 | 0 | 260 | * |
| | | Recyclables | 616 | 168 | 59 | 91 | 327 | *** |
| | **Disposed** | Within | 378 | 77 | 43 | 10 | 188 | |
| | | All others | 1400 | 115 | 57 | 0 | 260 | *** |
| | | Total | 784 | 90 | 55 | 0 | 260 | *** |
| | | Recyclables | 616 | 147 | 42 | 83 | 256 | *** |
| | **Recyclables** | Within | 231 | 40 | 43 | 1 | 129 | |
| | | All others | 1232 | 157 | 52 | 83 | 327 | *** |
| | | Total | 616 | 168 | 59 | 91 | 327 | *** |
| | | Disposed | 616 | 147 | 42 | 83 | 256 | *** |

t-test significance:

* = <0.05,

** = < 0.01,

*** = < 0.001

other comparisons had some distance between the two data points. The data within each waste type group tended to cluster (the distances within the groups were less than the distances outside the groups, statistically significant by t-tests on the means) but there was a clear overlap between the total waste and disposed waste streams although the small difference in the mean distances within and across the groups was significant (due to the large N in each group). Recyclables were clearly farther away from the total and disposed waste stream groups. The disposed waste stream data points were slightly farther away from the recyclables than the total waste stream data points were for the percentage data; the opposite was true for the rate data. The clusters determined for rates were slightly looser than those for the percentages, as differences in waste generation amplify differences in percentage data, and because the rate data were not constrained by having to sum to 100%.

Comparing the same borough distances across the years (i.e., Manhattan 1990 to Manhattan 2004-2013-2017, and Manhattan 2004 to Manhattan 2013–2017, and Manhattan 2013 to Manhattan 2017, and similarly for the other boroughs) showed the differences for the percentage data for the total waste stream were less than that for disposed waste and recyclables (Table 2). Comparing across boroughs in the same year (i.e., 1990 Manhattan to 1990 Bronx-Brooklyn-Queens-Staten Island, and similarly for 2004, 2013 and 2017) showed the difference in percentage data for recyclables was less than that for total waste and disposed waste streams. Overall, the distance between the percentage results for the same borough across different years was greater than the distance between results across boroughs in the same year, although the difference was not significant except in the case of recyclables. The rate data (Table 3) have greater differences, but that is because the percentage data were on one scale and the rate data are measured in kg/p/yr which had larger absolute values. For the rate data, the differences across years in the same borough were much less for recyclables data, and the total data varied slightly less than the disposed data. For the same year across different borough data, recyclables had much less difference than the disposed data, and the total data had the most differences. These distinctions were difficult to discern from the heat maps alone, although the rate data for recyclables were all bright red (lower triangle in Fig 6), indicating very little difference among the distances, which is quantitatively shown in Table 3. The differences between the same borough across years compared to the differences across boroughs for the same year was greater for almost all rate data for total waste and disposed waste but always less for recyclables data. The differences were significant overall for both the disposed data and the recyclables data.

The PCAs (Fig 7) support the heat map depictions. The first two axes of the PCAs captured nearly all the variance in the multidimensional array (~90%), meaning Fig 7 fairly completely depicted the distance relations. The total and disposed waste stream groups overlapped. Disposed wastes were farther from recyclables for percentages but closer for rate data, especially for the 2004-2013-2017 data sets. The recyclables formed a clearly defined separate cluster from the other two waste types. EPA data tended to be separate from NYC data, and there were evident distinctions between EPA total and disposed wastes. The borough data tended to cluster across years, more closely for 2004-2013-2017 than for 1990 data.

**Comparative sites.**   The heat map of the Euclidean distances between the pairs of the NYC and EPA and comparative data sets for disposed wastes by percentage (Fig 8) illustrated certain similarities and differences. Fig 8 was organized so that the same jurisdiction and then jurisdictions similar to it were grouped: the NYC data, then the New York State data sets (with Brookhaven NY and Onondaga NY groups), Washington (Clark County WA, King County WA, Pierce County WA, Seattle WA, Thurston County WA, and then the state-wide data), Oregon, California (the single locality results then the state-wide data sets), Orange County NC, the remaining single localities from the 2004 NYC study (Philadelphia PA, Pennsylvania,

**Table 2. Mean Euclidean distances for percentage data (across boroughs and years).**

|  | Comparison | Percentage (mean) | | |
|---|---|---|---|---|
|  |  | Total | Disposed | Recyclables |
| **Manhattan** | 1990–2017 | 8.4 | 8.5 | 10.6 |
| 1990 | Other 1990 | 6.6 | | |
| 2004 | Other 2004 | 12.5 | 12.0 | 8.9 |
| 2013 | Other 2013 | 15.8 | 15.9 | 9.8 |
| 2017 | Other 2017 | 12.9 | 12.0 | 12.4 |
| **Bronx** | 1990–2017 | 11.7 | 12.1 | 10.9 |
| 1990 | Other 1990 | 5.2 | | |
| 2004 | Other 2004 | 8.0 | 6.9 | 13.2 |
| 2013 | Other 2013 | 10.7 | 10.1 | 10.2 |
| 2017 | Other 2017 | 11.1 | 8.8 | 10.3 |
| **Brooklyn** | 1990–2017 | 8.9 | 10.3 | 9.1 |
| 1990 | Other 1990 | 6.0 | | |
| 2004 | Other 2004 | 6.7 | 6.6 | 6.4 |
| 2013 | Other 2013 | 8.6 | 8.9 | 6.1 |
| 2017 | Other 2017 | 8.2 | 7.5 | 7.3 |
| **Queens** | 1990–2017 | 9.6 | 12.5 | 11.0 |
| 1990 | Other 1990 | 6.3 | | |
| 2004 | Other 2004 | 6.3 | 6.6 | 5.9 |
| 2013 | Other 2013 | 9.0 | 9.8 | 7.1 |
| 2017 | Other 2017 | 8.3 | 9.2 | 6.9 |
| **Staten Island** | 1990–2017 | 10.8 | 11.6 | 12.7 |
| 1990 | Other 1990 | 8.0 | | |
| 2004 | Other 2004 | 8.6 | 9.9 | 7.3 |
| 2013 | Other 2013 | 12.1 | 13.3 | 6.0 |
| 2017 | Other 2017 | 11.4 | 12.3 | 7.5 |
| **Same borough** | **Across years** | 9.9 | 11.0 | 10.9 |
| **Same year** | **Across boroughs** | 9.1 | 9.1 | 8.4 |
| **t-test means test** |  | p = 0.25 | p = 0.08 | * |

t-test significance:

* = <0.05

Phoenix AZ, and Georgia), Missouri, Iowa, and finally EPA. Qualitatively, similar red-orange colors identified 2004-2023-2017 NYC, most of the Clark County WA data sets (although the 1993 data were not as similar to the 2012 data), Pierce County WA 2017-2018-2019 (single-family residential and multi-family residential), Seattle WA 2006-2010-2014, Oregon metro 1993-1998-2000-2002, and California state-wide 2008-2014-2018-2021 as self-similar data sets. Data sets that tended to be different from other data, as indicated by the most blue-purple areas, included 1988 and 1990 Seattle WA, Washington state-wide generally, Phoenix AZ (2003), Missouri generally, and 1990 Iowa. NYC, the focus of this investigation, was most similar to 2012 and 2014 Brookhaven NY, 1998 and 2005 Onondaga NY, 2016–2019 Pierce County WA, 1999 and 2004 Thurston County WA, and 2008, 2014, 2018, and 2021 California state-wide data. For the 2004-2013-2017 NYC data sets, Albany NY (2009), 2011, 2015 and 2019 King County WA, 2002, 2006, 2010, and 2014 Seattle WA, and the Oregon metro data sets were also similar. The 1990 NYC data set was similar to 1993, 1995, and 1999 Clark County WA, 1999 King County WA, 1995 Pierce County WA (multi-family residential), Philadelphia

**Table 3. Mean Euclidean distances for rate data (kg/p/yr) (across boroughs and years).**

| | Comparison | Rate (mean) | | |
|---|---|---|---|---|
| | | Total | Disposed | Recyclables |
| **Manhattan** | 1990–2017 | 58.8 | 70.8 | 6.2 |
| 1990 | Other 1990 | 44.9 | | |
| 2004 | Other 2004 | 53.2 | 45.4 | 11.7 |
| 2013 | Other 2013 | 55.2 | 49.1 | 12.1 |
| 2017 | Other 2017 | 44.7 | 37.4 | 12.6 |
| **Bronx** | 1990–2017 | 44.7 | 46.6 | 4.8 |
| 1990 | Other 1990 | 62.2 | | |
| 2004 | Other 2004 | 48.7 | 33.2 | 21.6 |
| 2013 | Other 2013 | 41.9 | 31.6 | 18.5 |
| 2017 | Other 2017 | 42.0 | 27.3 | 16.7 |
| **Brooklyn** | 1990–2017 | 58.8 | 68.7 | 4.5 |
| 1990 | Other 1990 | 43.8 | | |
| 2004 | Other 2004 | 37.6 | 30.3 | 12.7 |
| 2013 | Other 2013 | 34.2 | 29.0 | 10.7 |
| 2017 | Other 2017 | 32.0 | 25.1 | 9.8 |
| **Queens** | 1990–2017 | 52.5 | 64.5 | 8.8 |
| 1990 | Other 1990 | 43.3 | | |
| 2004 | Other 2004 | 36.4 | 30.3 | 12.2 |
| 2013 | Other 2013 | 35.8 | 31.7 | 10.7 |
| 2017 | Other 2017 | 32.8 | 27.3 | 9.7 |
| **Staten Island** | 1990–2017 | 69.0 | 79.2 | 10.3 |
| 1990 | Other 1990 | 64.6 | | |
| 2004 | Other 2004 | 72.3 | 62.0 | 21.9 |
| 2013 | Other 2013 | 62.3 | 51.3 | 21.3 |
| 2017 | Other 2017 | 58.4 | 47.2 | 17.4 |
| **Same borough** | **Across years** | 56.8 | 66.0 | 6.9 |
| **Same year** | **Across boroughs** | 47.3 | 40.9 | 14.6 |
| **t-test means test** | | p = 0.07 | ** | *** |

t-test significance:

* = <0.05;

** = <0.01;

*** = <0.001

PA (1999), Pennsylvania (2001), and 1997 and 2005 Iowa. The NYC data were most different from 1987, 2009, and 2016 Washington State, Phoenix AZ (2003), and 1990 Iowa.

Quantitatively, Table 4 lays out the closest and farthest comparison data sets, using a somewhat arbitrary standard that 10 is close and 20 is far away. The data sets closest to 1990 NYC, for instance, although they tended to be older reports, were not strictly before 2000, nor were they entirely East Coast, or strictly urban areas. The closest results to 2004 NYC included a number of New York data sets, but not many from the 2004-era, nor were they strictly urban results. The results closest to NYC 2013 and NYC 2017 tended to be more recent data sets and included some New York state results, but older and less urban results sets were represented. The two most similar results were the 2013 NYC and 2018 California residential waste streams: opposite coasts, one a particular jurisdiction and the other a state-wide study, although somewhat temporally close. The most dissimilar results to the 1990 NYC data sets included a

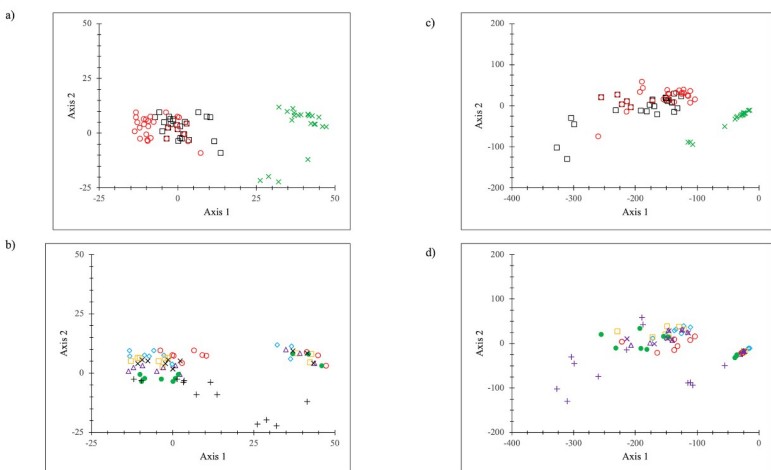

**Fig 7.** Boroughs and all NYC and EPA PCAs (a) percentages by waste type (b) percentages by location; (c) rates (kg/p/ yr) by waste type; (d) rates (kg/p/yr) by location Key for (a) and (c): □ = total waste stream (in black); o = disposed waste stream (in red); x = recyclables (in green) Key for (b) and (d): x = NYC (in black); o = Manhattan (in red); ◇ = Bronx (in blue); = Brooklyn (in orange); Δ = Queens (in purple); ● = Staten Island (in green); + = EPA (in black).

number of West Coast data sets, but not many of the most recent waste composition studies. NYC 2004 results were dissimilar from a number of West Coast studies and some of the very early data sets, which might be expected, but also several of the New York state studies. NYC 2013 was dissimilar from early West Coast studies and studies from the other data set; 2017 NYC was most dissimilar from earlier state-wide studies. The two studies most dissimilar from

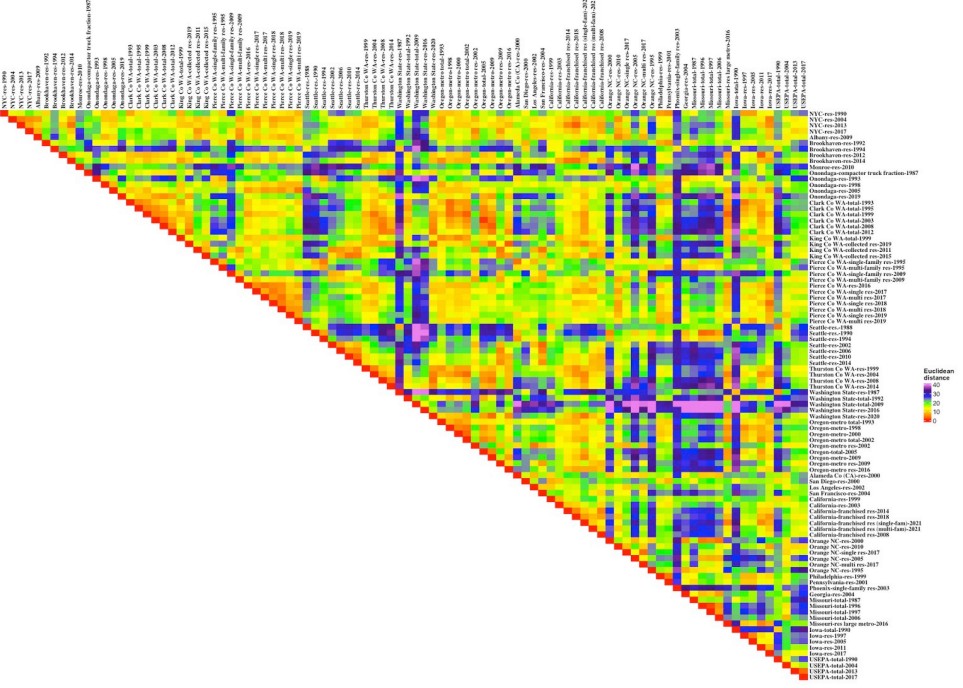

**Fig 8. Euclidean distances for the comparison sites organized by location (percentages).**

Table 4. **a.** NYC data sets: 12 nearest and farthest comparison data sets to NYC 1990 and NYC 2004. **b.** NYC data sets: 12 nearest and farthest comparison data sets to NYC 2013 and NYC 2017.

**a**

| | NYC 1990 | | NYC 2004 | |
|---|---|---|---|---|
| **Similar (d<10)** | **n = 12** | | **n = 25** | |
| | PA 2001 | 5.6 | NYC 2013 | 4.2 |
| | IA 1997 | 6.5 | IA 2017 | 4.7 |
| | IA 2005 | 6.5 | NYC 2017 | 5.5 |
| | Philadelphia PA1999 | 6.7 | Orange NC 2010 | 5.7 |
| | Clark Co WA 1993 | 7.2 | CA 2018 | 6.0 |
| | King Co WA 1999 | 8.1 | Albany NY 2009 | 6.3 |
| | Pierce Co WA multi. 2019 | 8.3 | Pierce Co WA single 2017 | 6.5 |
| | OR 1998 | 9.0 | Pierce Co WA single 2018 | 6.7 |
| | Onondaga NY 1998 | 9.3 | Brookhaven NY 2014 | 6.8 |
| | OR 1993 | 9.5 | OR 1993 | 7.4 |
| | Pierce Co WA multi. 1995 | 9.9 | Pierce Co WA single 2019 | 7.7 |
| | Onondaga NY 2005 | 9.9 | Pierce Co WA multi. 2018 | 7.8 |
| **Different (d>20)** | **n = 28** | | **n = 15** | |
| | Monroe Co. NY 2010 | 22.8 | MO 1997 | 21.6 |
| | EPA 2013 | 23.1 | EPA 1990 | 21.9 |
| | IA 1990 | 23.2 | Seattle WA 1988 | 22.0 |
| | Seattle WA 2002 | 23.7 | Brookhaven NY 1994 | 22.2 |
| | EPA 2017 | 24.4 | MO 1987 | 22.8 |
| | WA 1987 | 25.2 | Monroe Co. NY 2010 | 23.4 |
| | Seattle WA 2006 | 25.3 | WA 2016 | 23.8 |
| | Orange Co. NC 2005 | 25.9 | Orange Co. NC 1995 | 24.0 |
| | Pierce Co WA single 2009 | 28.2 | Phoenix AZ 2003 | 25.9 |
| | WA 2016 | 29.2 | WA 1987 | 26.4 |
| | WA 2009 | 29.2 | IA 1990 | 28.0 |
| | Phoenix AZ 2003 | 30.1 | WA 2009 | 28.7 |

**b**

| | NYC 2013 | | NYC 2017 | |
|---|---|---|---|---|
| **Similar (d<10)** | **n = 25** | | **n = 17** | |
| | CA 2018 | 2.7 | OR 2002 | 5.2 |
| | Brookhaven NY 2014 | 4.0 | NYC 2004 | 5.5 |
| | NYC 2004 | 4.2 | King Co WA 2011 | 6.0 |
| | CA 2014 | 5.2 | OR 2009 | 6.4 |
| | IA 2017 | 5.3 | Pierce Co WA multi. 2009 | 6.5 |
| | CA 2008 | 6.6 | NYC 2013 | 7.0 |
| | NYC 2017 | 7.0 | Orange Co. NC 2010 | 7.4 |
| | Thurston Co WA 1999 | 7.3 | CA 2008 | 7.5 |
| | Pierce Co WA single 2017 | 7.3 | Albany NY 2009 | 7.7 |
| | OR 1993 | 7.4 | CA 2018 | 8.1 |
| | Thurston Co WA 2004 | 7.6 | Los Angeles CA 2002 | 8.7 |
| | CA 2021 | 7.9 | King Co WA 2015 | 8.9 |

(*Continued*)

**Table 4.** (*Continued*)

| Different (d>20) | n = 19 | | n = 23 | |
|---|---|---|---|---|
| | EPA 1990 | 23.2 | EPA 1990 | 22.0 |
| | Seattle WA 1988 | 23.4 | MO 1997 | 22.3 |
| | Seattle WA 1990 | 23.6 | WA 1992 | 22.4 |
| | MO 1987 | 24.4 | Brookhaven NY 1994 | 22.8 |
| | MO 1996 | 24.4 | WA 2016 | 23.8 |
| | Phoenix AZ 2003 | 24.5 | Phoenix AZ 2003 | 24.5 |
| | MO 1997 | 25.3 | MO 1987 | 26.0 |
| | WA 2009 | 25.4 | Monroe Co. NY 2010 | 26.1 |
| | Orange Co. NC 2005 | 25.4 | WA 1987 | 26.7 |
| | Orange Co. NC 1995 | 26.9 | Orange Co. NC 1995 | 27.0 |
| | WA 1987 | 27.0 | IA 1990 | 29.4 |
| | IA 1990 | 29.0 | WA 2009 | 30.7 |

(n indicates the number of results meeting the indicated standard)

the NYC data sets were the 2009 Washington residential data, and the Monroe County NY (2010) residential data (Monroe County is the jurisdiction containing the city of Rochester).

A second heat map (Fig 9) organized the data by time. Table 2 showed that the sampling year across-borough data were closer than the same borough measured across the sampling events data, although the difference was not significant. An analogous result here would have showed blocks of red and orange data along the diagonal. But that was not the case; there was one small block associated with some 1998–1999 data sets, which were all from the West

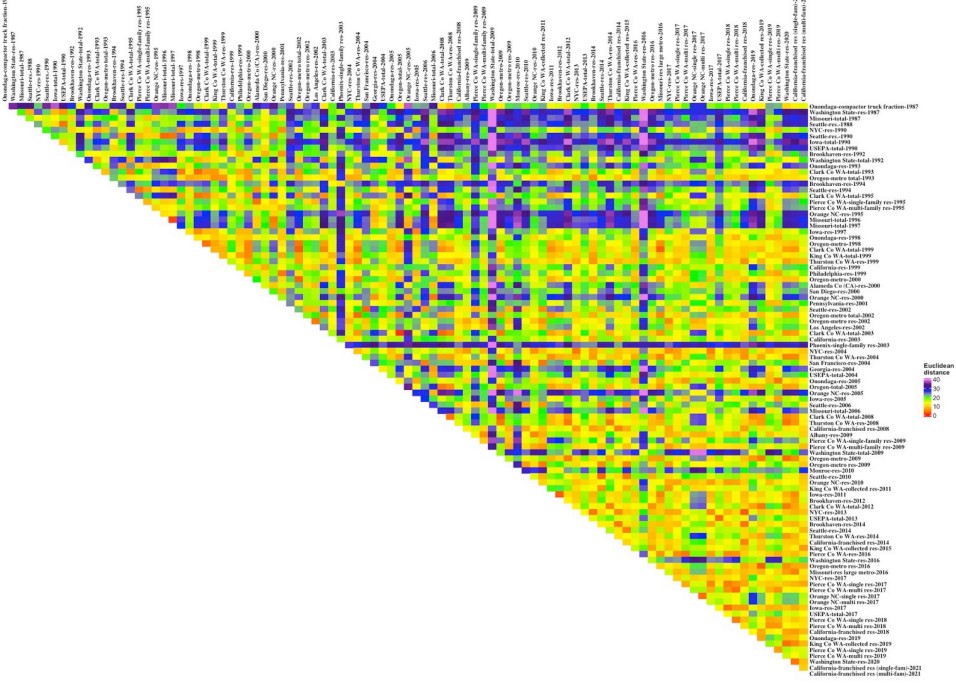

**Fig 9. Euclidean distances for the comparison sites organized by time (percentages).**

**Table 5. Comparison site multivariate distance summaries.**

| All | N | Mean | SD | Max. | Min. |
|---|---|---|---|---|---|
| | **4560** | **18** | **8** | **50** | **1** |
| By years | | | | | |
| pre-2010 | 1830 | 19 | 8 | 50 | 1 |
| 2010+ | 595 | 14 | 6 | 34 | 3 |
| By jurisdiction | | | | | |
| NYC | 6 | 11 | 6 | 19 | 4 |
| Brookhaven | 6 | 16 | 4 | 22 | 9 |
| Onondaga | 10 | 19 | 8 | 32 | 8 |
| All NY (not NYC) | 55 | 19 | 7 | 33 | 8 |
| **Clark Co** | 15 | 8 | 3 | 15 | 3 |
| King Co | 6 | 13 | 4 | 17 | 8 |
| Pierce Co | 55 | 12 | 6 | 26 | 4 |
| Seattle | 21 | 18 | 10 | 30 | 3 |
| **Thurston Co** | 6 | 8 | 3 | 11 | 4 |
| WA State | 10 | 25 | 12 | 47 | 13 |
| All WA | 666 | 18 | 8 | 47 | 3 |
| OR | 36 | 12 | 5 | 22 | 3 |
| CA State | 21 | 11 | 5 | 21 | 5 |
| All CA | 55 | 14 | 5 | 24 | 5 |
| Orange Co | 15 | 15 | 6 | 24 | 4 |
| MO | 10 | 17 | 10 | 29 | 2 |
| IA | 10 | 18 | 10 | 30 | 5 |
| All others | 190 | 19 | 8 | 37 | 2 |
| EPA | 6 | 16 | 8 | 26 | 3 |

**Bold** indicates mean < (all data mean-SD)

Coast, too, but the sections of similar data in this figure were to be found in the interior of the diagram, where the sampling was not especially co-temporal. There is the suggestion that the lower third of the diagram (results from 2010 or later) are "yellower" than the remainder of the heat map, meaning that they are closer to each other.

Analytical comparisons of the within-group distances were made for some temporal and geographical groupings (Table 5). The mean distance across the 4,560 comparisons was 18, with a standard deviation of 8. The post-2010 data had a within-group mean distance less than that (14). NYC data had a mean within-group distance of 11. Pierce County (WA), King County (WA), Oregon, and California state-wide data sets all had similar distance measures. Notably, the only two jurisdictions where the mean within-group distances were less than a standard deviation from the overall mean were Clark County (WA) and Thurston County (WA).

The PCA of the comparison data (Fig 10) (the first two axes account for 76.4% of the data) tends to confirm the general lack of geographic or temporal organization to the 96 data sets. There was no overt clustering in the representation, unlike the PCA of the boroughs-all NYC-EPA data. The PCA suggested that NYC data were "representative" waste data, given their central location in the data cloud; but note that EPA data were not as central (Fig 10a). The earliest data sets (pre-1995) were somewhat different from the post-1995 data (Fig 10b), but it is not a well-organized clustering array. Mapping the data sets in broad geographical

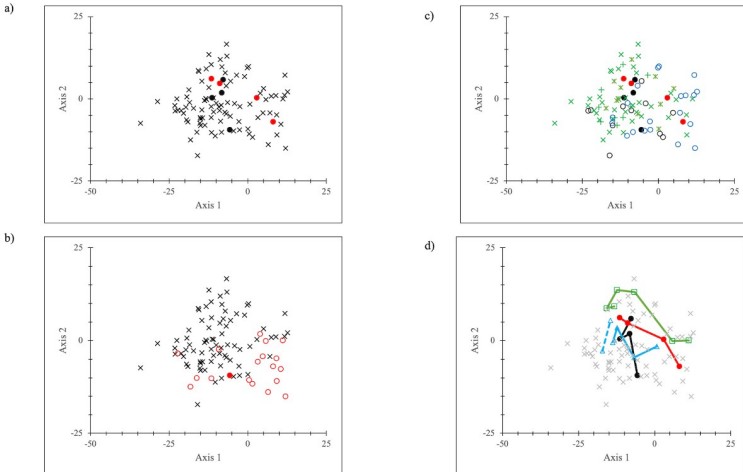

**Fig 10.** PCA of NYC-EPA-comparison disposal data a: ● = NYC studies (in black), ● = EPA data (in red), x = comparison data (in black); b: ● = 1990 NYC (in black), ○ = 1995 and earlier data (in red), x = post 1995 data (in black); c: ● = NYC data (in black), ● = EPA data (in red), x = Washington state and jurisdictions data (in green), + = Oregon data (in green), * = California state and jurisdictions data (in green), ○ = other data (in blue); d. black = NYC data, red = EPA data, green = Seattle data, blue = California data, x = all other data.

groups (Fig 10c) also did not show any meaningful clustering, although the West Coast data sets tended to be to the left and top of the data and the other data tended to be down and to the right. In Fig 10d, a mapping of the NYC, EPA, Seattle WA, and California data (the dotted line connected the two 2021 California data sets, single-family and multi-family residential), with lines added to indicate temporal progression (1990 NYC, 1990 EPA, 1988 Seattle WA, and 1999 California data were down and to the right along the jurisdiction lines) suggested that for these particular data sets, there was some order to change over time. Changes in the waste streams over several decades had some degree of common direction across multiple variables.

## Discussion

This paper analyzed NYC waste composition data from four studies (1990, 2004, 2013, 2017) using conventional analytical approaches and introduced multivariate distance comparisons as a means of understanding waste composition data. Simplifying comprehensive waste characterizations to paper, plastic, glass, metal, food, and yard waste (and balance) categories enabled comparisons to be made across the four NYC waste characterizations, corresponding EPA data sets, and selected other waste characterizations from other jurisdictions.

The findings based on conventional single parameter analyses were that NYC waste composition from 1990 to 2017 was similar to disposal percentages reported from other jurisdictions and tended to differ from EPA data, especially when measured as rates. It is not surprising that NYC data, reported only for residential wastes, differed from EPA data, which is based on the entire waste stream, especially in terms of the amounts of specific materials. However, trends in how the data changed, such as whether materials increased or decreased over time, and the degree of change over time were also different between NYC and EPA data, suggesting that they are not reporting the same waste phenomena.

Food waste, as a percentage of the NYC waste stream, increased over time. Because the overall amount of waste managed in NYC decreased over the 1990–2017 time period [38], although overall percentages of food waste generation increased from 1990–2017, food waste

generation rates actually decreased from 2004 to 2017 for NYC as a whole. The same was true for food waste disposal. As with the NYC percentage data, EPA percentage data increased from 1990 to 2017. Unlike the NYC rate data, EPA rate data also showed increases in food waste generation and disposal. The comparison data sets for food waste disposal percentages were in accord with the NYC and EPA percentage increases.

Thyberg et al. [17], in a meta-analysis of 62 waste characterization studies from 1987–2013, reported that disposed waste sorts had lower percentages of food waste than EPA data, and that food waste disposal percentages were increasing over time. They also reported not-significant increases in food disposal rates over time from the waste sort data. The percentage for food waste in the disposed waste stream was 14.7%, which translated to approximately 102 kg/p/yr. This is very similar to EPA data for 2013 for disposed food rates, although a much smaller percentage of food waste than EPA reported. The data we collected for food waste disposal for the comparison data set also found increasing percentages of disposed food waste. The regression line generated from the comparison data sets, the NYC data, and EPA data all suggested food waste in 2013 was about 20% of the disposed waste stream. The NYC rate data suggested residential food waste in 2017 was slightly less than it was in 2004, and the NYC rate data were about 40% lower than the Thyberg and EPA rate data. NYC data are for residential wastes only, and therefore exclude large amounts of food waste generated at supermarkets, schools, and restaurants, and smaller amounts generated at offices and other businesses. This may account for the smaller rate data for NYC compared to Thyberg et al. and EPA.

Kumar and Garg [20] used 27 waste composition studies from 2001 through 2018. They extracted the same six waste categories as we used, but then normalized the data sets instead of incorporating the balance of waste into the calculations. They did the same for the EPA data. Since they normalized to the sum of the six categories, the percentages produced in their paper were not comparable to those we have reported. Fig 5 of their paper included estimates of disposed rates for food, plastics, and paper. Food waste rates were set at 11% greater than EPA data for 2017. 2017 EPA data were about 105 kg/p/yr, making the Kumar and Garg estimate about 115 kg/p/yr. This value was 75% greater than the NYC rate data for 2017.

As with food, plastics percentages increased from 1990–2017 in the NYC total waste stream; however, partly because of recycling, the disposal percentages only increased by 13% across NYC as a whole and decreased slightly from 2004 to 2017. Plastic recyclables as a percentage of the recyclables stream, recovery percentages, and the amount of plastics collected for recycling all increased 2004–2017. The increases were much more substantial than reported by EPA. EPA data also showed increases in plastics percentages and rates for the total waste stream and the disposed wastes stream. The NYC plastics percentage data increased, but not the disposal rate data. Again, the comparison data sets for plastics disposal percentages tend to conform much more with the NYC data than with EPA data.

Milbrandt et al. [31] analyzed plastics composition data from 44 waste composition studies, nearly all conducted between 2010 and 2019. They asserted that waste composition was essentially consistent over this time period. Plastic waste was found to be 13.7% of disposed waste for 2019. We translated their estimate of 44 MT nationwide for disposed plastics for 2019 to approximately 120 kg/p/yr, about 25% greater than EPA's value for 2017 and three times the NYC waste stream rate estimate. The NYC data estimate plastic disposed waste at 14.3% for 2017; the regression trend for the comparison data modeled plastic waste at 15.2% for 2019 (the year used by Milbrandt et al.), and those data suggested a significantly increasing trend for plastics over time, although Milbrandt et al. in order to include waste composition studies over a range of years made the assumption that the waste stream composition was consistent over the decade of the 2010s. Kumar and Garg [20] found plastics were 14% greater than EPA data

in 2017. EPA disposal in 2017 was 90 kg/p/yr, making the Kumar and Garg estimate approximately 105 kg/p/yr. This was approximately 2.5 times the size of the 2017 NYC disposed plastics waste stream rate.

NYC data showed paper declined over the nearly three-decade sampling period except in Manhattan as a percentage of the waste stream and disposal stream, very steeply so for disposal rates. The per capita amount of paper recycling held steady; the paper recovery rate increased. Similar patterns were found for the comparison sites, including EPA. EPA recovery rates were much higher than NYC data. NYC and comparison waste composition study percentages of disposed waste diverged from EPA data after 2005, with NYC and the linear regression line values increasingly larger than the EPA data. Paper disposal rates, as calculated by EPA, were only 20% greater than NYC disposal rates (64.1 kg/p/yr compared to 55.5 kg/p/yr) although NYC data only include residential wastes and EPA wastes include businesses and institutional waste as well as residential waste.

Milbrandt et al. [19] used 52 waste composition studies, all but three from 2012–2020, to estimate paper waste composition. They assumed waste composition was consistent from 2015–2020, although the linear trend we generated for disposed paper showed a general decreasing trend for 1987–2021. They found paper was 26% of total wastes in 2019 at 109,769,000 tons. We translated that to an annual waste generation rate of approximately 300 kg/p/yr. That was 50% greater than EPA's rate for 2017, and approximately 3.5 times greater than the calculated NYC total waste stream rate for paper, although the NYC total waste stream percentage for paper and the EPA value for paper were both near 26% for 2017. Milbrandt et al. calculated disposed paper at 68,066,000 tons, which we translated to 190 kg/p/yr. That was approximately three times the EPA and NYC disposal rates for 2017. Milbrandt et al. did not calculate the percentage of paper in the disposed waste stream. Kumar and Garg [20] determined that disposed paper was 260% of the EPA datum for 2017 (65 kg/p/yr). Their estimate for paper waste disposal was therefore on the order of 150 kg/p/yr, and approximately three times the NYC disposal value of 55.5 kg/p/yr.

Thyberg et al. [17], Milbrandt et al. [31], Milbrandt et al. [19] and Kumar and Garg [20] suggested national data sets have much larger amounts of food, plastics, and paper waste than the NYC waste streams. In part this was because the NYC waste composition studies were of the residential waste stream, not the entire waste stream. However, part of the differences also were due to the shrinkage of the NYC waste stream from 1990–2017 (data presented in [38], see also S6 File). A major driver of the change seems to be decreases in paper waste generation, approximately 40% since 1990. However, plastics waste generation decreased in NYC by approximately 15% from 2004, and even food waste generation decreased very slightly across all of NYC from 2004. Percentage waste disposal data from the comparison sites also showed declining paper disposal and increasing percentages of plastics and food–true also for the NYC data sets.

Our selection of comparisons seemed to show that the NYC waste characterization data for food and especially plastics and paper in the disposed waste stream approximately matched national values and trends. The NYC data lie close to the regression lines generated from the waste sort data sets. Plastics and paper trends for EPA data do not fit the linear regression lines. Therefore, not only were NYC disposal percentages different from EPA data, but the selection of other waste characterization data sets also tended not to agree with EPA data trends for plastic and paper.

Recycling has usually been found to be greater in California, Oregon, and Washington, and the comparison data presented here can be interpreted as supporting that. Materials that were targets of recycling programs such as plastics and paper were found at lower percentages relative to the other data sets for the West Coast data sets. Removing recyclables from the waste

stream, which can be assumed to decrease the waste stream size, meant that equivalent amounts of food waste would be proportionately greater percentages in diminished-size waste streams, and this appeared to be reflected in the data. Note there was no information presented regarding waste stream size for the comparison data sets, nor about absolute rates of disposal. The opposite effect of increased recycling could have held for the "other" data sets: less recycling meant relatively higher percentages of paper and plastic in the disposed waste stream, and relatively smaller percentages of food waste.

Multivariate data analyses based on distance measurements provide a different perspective on the data sets. These analyses allow for wholistic comparisons of waste composition data sets. Instead of balancing differences across a range of parameters, the multivariate distance comparisons enable statements about system differences: for instance, wastes are different in Manhattan compared to Brooklyn, or to Onondaga County (NY), or in the state of California. The distance data generate quantities comparing the data sets, and these quantities can be qualitatively and analytically compared.

For instance, in NYC overall recyclables waste composition was found to be very different from disposal and total waste stream compositions. This is not a surprising finding and in fact seems to be an obvious and unremarkable result. Recyclables separations are intended to be very different from the total waste stream: they are a selection from that larger set, and many elements of MSW are not supposed to be in the recyclables. However, in the simplified NYC waste stream depiction, four of the specified elements (paper, plastics, glass, and metals) are in part targeted recyclables, and yard waste, one of the other two elements, is not a major element of urban waste streams. Nonetheless, the overall nature of recyclables setouts is distinct from the overall waste stream and the disposed waste stream.

The multivariate distance measures can generate insight into overall changes in the waste stream over time and space. The percentage data suggest that generally waste composition is more similar across the boroughs in one year than it is in the same borough across the four different waste characterizations, except for Manhattan where that is only true for the 1990 waste data set. The Manhattan data sets may be outliers partly because the percentage and amounts of paper and plastics in the waste stream did not change as much in that borough as they did in the other boroughs. The change in waste composition for individual boroughs for percentages for individual constituents could be very large over the 28-year study period (Table 6), and were often substantial for 2004–2017 (Table 7). The degree of change is greater than the differences across the sample data for particular study years (Table 8).

The rate data also indicate that waste composition and disposal are more similar across boroughs in the same year compared to the same borough across the different sorting events. The change in waste composition for individual boroughs for percentages for individual constituents could be very large over the 28-year study period (Table 6). The degree of change is greater than the differences across the sample data for particular study years for total waste stream and disposal data (Table 7). This was not true for recyclables where recyclables composition is more similar for the same borough across the 2004-2013-2017 sorts, and also that recyclables rate data are much more similar than the total waste stream or disposal data. The change in recyclables data for paper over the 28-year period was notably small, and until the 2010s paper was by far the dominant recyclable in recyclable setouts.

Overall, this suggests that changes generated by social and economic changes over time have more control on waste composition and disposal compared to socio-economic and land use differences across the five boroughs. The data suggest that when considered in terms of the setout amounts, recycling habits are largely similar across the NYC as a whole and tended to change more similarly over time.

**Table 6. Change in waste sort parameters, total and disposed waste streams 1990–2017 (percentages).**

| | Percentage | | | Rate | | |
|---|---|---|---|---|---|---|
| | **Food** | **Plastics** | **Paper** | **Food** | **Plastics** | **Paper** |
| **Manhattan** | | | | | | |
| Total | +45 | +51 | +1 | -7 | -3 | -36 |
| Disposed | +87 | +54 | -19 | -8 | -24 | -60 |
| **Bronx** | | | | | | |
| Total | +94 | +76 | -27 | +60 | +45 | -40 |
| Disposed | +94 | +67 | -40 | +36 | +17 | -58 |
| **Brooklyn** | | | | | | |
| Total | +60 | +76 | -11 | +13 | +24 | -37 |
| Disposed | +91 | +72 | -30 | +11 | 0 | -59 |
| **Queens** | | | | | | |
| Total | +78 | +60 | -24 | +34 | +20 | -43 |
| Disposed | +115 | +44 | -41 | +31 | -13 | -64 |
| **Staten Island** | | | | | | |
| Total | +58 | +86 | -6 | +27 | +50 | -24 |
| Disposed | +97 | +71 | -27 | +27 | +10 | -53 |

The comparative data sets for disposed wastes did not support either thesis very well. Data that were most similar to the NYC data sets were not either temporally or geographically similar, and the results that were most different did not necessarily occur at different times or in far-away or rural settings. There were much more self-similar results for jurisdictions than for

**Table 7. Change in waste sort parameters 2004–2017 (percentages).**

| | Percentage | | | Rate | | |
|---|---|---|---|---|---|---|
| | **Food** | **Plastics** | **Paper** | **Food** | **Plastics** | **Paper** |
| **Manhattan** | | | | | | |
| Total | +21 | +7 | -11 | 0 | -11 | -26 |
| Disposed | +23 | -2 | -10 | -6 | -25 | -32 |
| Recyclables | | +32 | +76 | | +112 | -2 |
| **Bronx** | | | | | | |
| Total | +30 | +13 | -14 | +14 | -2 | -25 |
| Disposed | +13 | +4 | -17 | -5 | -13 | -30 |
| Recyclables | | +24 | +79 | | +109 | -1 |
| **Brooklyn** | | | | | | |
| Total | +8 | +9 | -6 | -7 | -6 | -18 |
| Disposed | +8 | +2 | -4 | -10 | -16 | -20 |
| Recyclables | | +16 | +52 | | +72 | -4 |
| **Queens** | | | | | | |
| Total | +28 | +2 | -15 | +7 | -15 | -29 |
| Disposed | +27 | -13 | -13 | +3 | -29 | -29 |
| Recyclables | | +3 | +90 | | +80 | -26 |
| **Staten Island** | | | | | | |
| Total | +15 | +6 | -9 | -5 | -13 | -25 |
| Disposed | +15 | -8 | +1 | -10 | -28 | -21 |
| Recyclables | | +8 | +65 | | +92 | -18 |

**Table 8. Range of waste sort parameters (percentages) (measured over all five boroughs).**

| | Percentage | | | Rate | | |
|---|---|---|---|---|---|---|
| | **Food** | **Plastics** | **Paper** | **Food** | **Plastics** | **Paper** |
| **1990** | | | | | | |
| Total-Disposed | 21% | 33% | 14% | 20% | 27% | 28% |
| **2004** | | | | | | |
| Total | 27% | 19% | 31% | 26% | 18% | 42% |
| Disposed | 21% | 21% | 31% | 25% | 17% | 26% |
| Recyclables | | 31% | 33% | | 40% | 63% |
| **2014** | | | | | | |
| Total | 19% | 23% | 37% | 28% | 17% | 42% |
| Disposed | 16% | 31% | 40% | 26% | 17% | 30% |
| Recyclables | | 47% | 34% | | 49% | 61% |
| **2017** | | | | | | |
| Total | 36% | 24% | 33% | 28% | 18% | 42% |
| Disposed | 20% | 26% | 30% | 21% | 22% | 34% |
| Recyclables | | 27% | 21% | | 34% | 55% |

sample data sets from the same time period, which implies there is either a programmatic or potentially demographic factor to the composition of disposed wastes. The degree of similarity for most jurisdictions was not great, however (except for two counties in Washington). There is a suggestion that there is some overall similarity in waste composition for the studies collected from the 2010s, which was an assumption made by Milbrandt et al. [31]. The degree of similarity was not great, just relatively more similar compared to the overall degree of variation in the percentage data from 1987 to 2021. Using trends in PCA graphs suggested that restricting focus to just a few systems supported some similarity in the degree and overall direction of change in the overall waste stream across time. So, although the mapping of 96 separate results did not generate any kind of temporal order, selection of a few exemplars implied that there may have been broader social change occurring across the country that helped drive disposed waste composition over time. This could have been changes in underlying waste generation patterns such as less paper being generated, but more food and more plastics. Or it may have been that there were changes in waste management that were common to these jurisdictions, such as increased recycling and composting, for instance causing similarities in the changes in disposed wastes composition.

Multivariate distance measures generated information that expanded understanding of waste composition beyond that available from basic single parameter analyses. The single parameter analyses are very precise, but tend to form a cloud of information that is sometimes conflicting. So, when considering percentage data sets, when one percentage increases then necessarily another must decrease. Changes across rate data sets can be more uniform. However, the compilation of many parameter examples can hide simple overall conclusions due to the number of comparisons that are made. We discussed only three parameters (food, plastics, and paper) but the accumulation of detail between percentages, rates, total waste stream, disposed wastes, and recyclables made for formidable demands on mental bookkeeping. Considering distance relationships across sample sets does not necessarily reduce data complexity: the triangular matrices of the distance comparisons include many data points. Qualitative visualization means such as heat maps allow for determinations of data clusters, and these clusters may signify important similarities and differences at the system level. Because the distance measures are quantitative, analytical comparisons can also be made that refine understanding

of the similarities and differences. PCA is a useful tool to add visualization information to the data relations, serving as a means of confirming the quantitative and qualitative findings. Extraction of data from the matrices is time-consuming and very tedious; we are developing some Excel macros (see S7 File) that reduce the work somewhat.

## Conclusions

Multivariate distance comparisons created wholistic understandings of waste composition and led to some obvious, simplistic observations, such as disposed wastes are very different in composition from collected recyclables. The distance data supported the idea that waste composition at different locations at the same time is more similar than waste composition at the same site compared years and even decades apart. However, there was also some evidence that the changes in the disposed waste stream could be similar in nature for particular waste stream depictions. Using the multivariate data sets may be a means of describing wider trends in waste composition over time and space, compared to single variable analyses that were harder to discuss in broad terms.

It is clear that combining multivariate and standard analyses may improve our understanding of waste composition. The multivariate analyses can show broad patterns of similarity and differences for wastes as a whole based on the principle that points close to each other are similar and points farther way are different. Standard parameter comparisons can help illuminate exactly why the results are different or similar. Thus, disposal processes generate wastes that are different in overall character from recyclables separation. The way the overall character of disposed waste changes over time and distance is different from how recyclables change in character of time and distance. The multivariate analyses confirm that EPA modelled composition data are different in kind from composition results generated by physically sorting MSW.

Multivariate analyses can also offer the opportunity for differences in perspective. Standard reporting of waste composition work is often a long string of comparisons of individual materials results, either to past results or some standard, especially to EPA data. This can be very valuable if tedious to read. The lists of similar results in the multivariate sections of this paper are also difficult to process. However, the individual parameter analytical reports may give the impression that MSW is discrete piles of individual materials instead of a disordered conglomerate. Recovering 20% food waste seems feasible if the understanding is that the food waste is a discrete element of the waste stream, instead of being a mixed-in fifth of a large pile of otherwise indiscriminate materials. Looking for similarities and differences in waste samples in a multivariate mode does not assume that the materials comprising the waste composition are separate in any meaningful way. It actually requires great effort, either manually by homeowners or through some set of technological tools at various waste operations, to create separated wastes. Standard reporting on waste composition studies does not make that inherently clear.

Introduction of more wholistic waste stream assessments has the potential for immediate benefits in waste data studies and public planning. Clear identification of similar and different waste composition across jurisdictions adds to understanding of how programmatic factors such as different approaches to collection (Pay-as-You-Throw, single stream recycling) affect the basic units of waste. Being able to quantify the similarities and differences means that more sophisticated determinations of differences across waste systems can be considered. In New York State, where we have a project to assess the waste stream, state planners have created a tool to estimate waste composition based on simple land use classifications (rural, suburban, urban). We can use the wholistic assessments of waste composition to determine the degree that land use helps define waste composition and how other factors may be important, such as

basic programmatic differences, or demographics, or different approaches to public education. This may help focus and prioritize state efforts to achieve a more circular economy.

## Supporting information

**S1 File. New York City and EPA file use.**
(DOCX)

**S2 File. Waste characterizations in this study.**
(DOCX)

**S3 File. Notes on waste characterizations.**
(DOCX)

**S4 File. EPA data sets.**
(DOCX)

**S5 File. Comparisons of Euclidean and Manhattan distances for the NYC data.**
(DOCX)

**S6 File. Trend graphs for NYC waste generation and disposal rates and EPA disposal rates (reported in [38]).**
(DOCX)

**S7 File. Excel macros useful to extract triangular matrix data.**
(DOCX)

## Acknowledgments

The opinions, findings, and or interpretations of data contained herein are the responsibility of the University and do not necessarily represent the opinions, interpretations or policy of the New York State Department of Environmental Conservation.

Assistance from Samantha MacBride (formerly of DSNY) and Freda Carmelo (DSNY), Maria Bianchetti (OCRRA), and Luann Meyer (formerly of Monroe County Department of Environmental Services) with their respective composition studies is gratefully acknowledged. The comments offered by three PLOS One reviewers (Ashootosh Mandpe, Jude A. Okolie, and one anonymous reviewer) resulted in revisions that strengthened the paper, and so their efforts are gratefully acknowledged.

DJT drafted the paper; YW compiled most of seven-parameter data sets and generated most of the single parameter comparisons (DJT helped); DJT drew the graphs; YW conducted the PCAs; FF created the distance measures; SM did some statistics on the multi-variate data sets; MJ compiled the EPA data sets; GW assisted with data management; DJT and EH arranged for project funding; DJT, EH, and KLT provided project management; YW, FF, KLT, and EH edited the paper and helped shape the theses.

This is Waste Data and Analysis Center Contribution #36.

## Author Contributions

**Conceptualization:** David J. Tonjes, Yiyi Wang.

**Data curation:** David J. Tonjes, Yiyi Wang, Firman Firmansyah, Matthew Johnston.

**Formal analysis:** David J. Tonjes, Yiyi Wang, Firman Firmansyah, Sameena Manzur, Griffin Walker.

**Funding acquisition:** David J. Tonjes, Elizabeth Hewitt.

**Investigation:** David J. Tonjes, Yiyi Wang.

**Methodology:** David J. Tonjes, Yiyi Wang, Firman Firmansyah.

**Project administration:** David J. Tonjes, Krista L. Thyberg, Elizabeth Hewitt.

**Resources:** David J. Tonjes, Firman Firmansyah.

**Supervision:** David J. Tonjes.

**Visualization:** David J. Tonjes, Yiyi Wang, Firman Firmansyah.

**Writing – original draft:** David J. Tonjes.

**Writing – review & editing:** Krista L. Thyberg, Elizabeth Hewitt.

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
