## [Decision Letter · Decision Letter 0]

28 May 2024

PONE-D-24-14194Determining differences in waste generation using machine learning techniquesPLOS ONE

Dear Dr. Tonjes,

Thank you for submitting your manuscript to PLOS ONE. After careful consideration, we feel that it has merit but does not fully meet PLOS ONE’s publication criteria as it currently stands. Therefore, we invite you to submit a revised version of the manuscript that addresses the points raised during the review process.

We look forward to receiving your revised manuscript.

Kind regards,

Basant Giri, Ph.D.

Academic Editor

PLOS ONE

“Funding for some of this research was provided through the New York State Environmental Protection Fund as administered by the New York State Department of Environmental Conservation through a Memorandum of Understanding with Stony Brook University (AM 11643) effective September 11, 2019. YW, FF, SM, MJ, and KLT were funded by the MOU and DJT and EH were partially funded. The opinions, findings, and or interpretations of data contained therein are the responsibility of the University and do not necessarily represent the opinions, interpretations or policy of the New York State Department of Environmental Conservation.

Assistance from Samantha MacBride (formerly of DSNY) and Freda Carmelo (DSNY), Maria Bianchetti (OCRRA), and Luann Meyer (formerly of Monroe County Department of Environmental Services) with their respective composition studies is gratefully acknowledged.

DJT drafted the paper; YW compiled most of seven-parameter data sets and generated most of the single parameter comparisons (DJT helped); DJT drew the graphs; YW conducted the PCAs; FF created the distance measures; SM did some statistics on the multi-variate data sets; MJ compiled the EPA data sets; DJT and EH arranged for project funding; DJT, EH, and KLT provided project management; YW, FF, KLT, and EH edited the paper and helped shape the theses. 

This is Waste Data and Analysis Center Contribution #36.”

“DJT, YW, FF, SM, MJ, KLT, and EH were funded by New York State Environmental Protection Fund as administered by the New York State Department of Environmental Conservation through a Memorandum of Understanding with Stony Brook University (AM 11643) effective September 11, 2019. The sponsor played no riole in the study design, data collection and analysis, decision to publish or preparation of the manuscript.”

Reviewers' comments:

Reviewer's Responses to Questions

**Comments to the Author**

1. Is the manuscript technically sound, and do the data support the conclusions?

Reviewer #1: Partly

Reviewer #2: Yes

Reviewer #3: Yes

2. Has the statistical analysis been performed appropriately and rigorously? 

Reviewer #1: N/A

Reviewer #2: No

Reviewer #3: Yes

3. Have the authors made all data underlying the findings in their manuscript fully available?

Reviewer #1: Yes

Reviewer #2: Yes

Reviewer #3: Yes

4. Is the manuscript presented in an intelligible fashion and written in standard English?

Reviewer #1: No

Reviewer #2: Yes

Reviewer #3: No

5. Review Comments to the Author

Reviewer #1: Manuscript No.: PONE-D-24-14194

Thank you for the manuscript draft titled “Determining differences in waste generation using machine learning techniques”.

Comments on this article are as follows:

General Comments:

• Authors are advised to use common English (modern English) words that will be easy for readers to understand. Writing should be clear, concise and should be well-written in academic tone.

• Uniformity should be maintained throughout the manuscript regarding font size and style, uppercase and lowercase.

Title:

• Rewrite the title. It should be more clear and informative.

• Determining the “spatiotemporal” differences in waste generation “practices” using machine learning techniques. • • It's just a suggestion; authors may opt for a different title if they wish..

Abstract:

• Rewrite the abstract in a more elaborative way.

• Explain how the study was done, including any model organisms used, without methodological detail.

• Summarize the most important results and their significance.

• The abstract should not exceed 300 words.

Keywords:

• Go for more suitable keywords related to your manuscript.

• Each keyword should be self-explanatory. Multivariate, Disposal should be replace.

• Waste studies should be written as waste composition studies.

• Correct the formatical mistake of using lower/uppercase letters in writing keywords.

Introduction:

• Sentences should be improved for clarity & conciseness

o Line no 26 sentences written in brackets should come in normal sentence construction.

o Rewrite line no 27, “many & increasing.”

o Rewrite line no 28 “greatest current concern.”

• Authors are advised to write some examples/instances from their cited report/paper in their drafted manuscript. Readers didn’t read the whole paper (cited) just to get the reference of 2 lines written in your manuscript.

• Long sentences are written in brackets. Sentence construction should be done so that these bracketed sentences come in normal sentences.

• Last 3-4 paragraphs of the Introduction section should come under the heading of Materials and Methods

• The Introduction section ends with a short para that gives a glimpse of upcoming sections. “Conclude with a brief statement of the overall aim of the work and a comment about whether that aim was achieved.”

Materials and Methods:

• Provide a justification for the sentence/comment in line no 153.

• Line no 172 Elaborate what are those two exceptions

Discussion:

• Ambiguity is in line no 617. Correct that.

• Cite the journal/report you are referring to for comparison with your results. Line no 625.

Conclusion:

• Conclusion should be aligned with the objectives of the experiment/paper defined at the end of the introduction section.

• Rewrite line no 749; in academic writing, how can we write words like “Mind you”? Authors are advised to stick with academic tone of writing throughout the paper.

• Conclusion should end with a small paragraph depicting the application of the study and its future scope. Authors can showcase how their study can benefit the real world in the near future.

#Minor Revision

Reviewer #2: The study presents a new approach for waste composition analysis. A topic of utmost importance considering the issue of environmental sustainability, waste disposal and circular economy. The abstract is generic and should be improved to include study novelty, motivation and quantitative information.

Additionally the following suggestions are proposed:

Interconnection between MSW composition and treatment/disposal methods as well as countries legislation should be discussed in the introduction

It is not clear whether the EPA dataset is based on a particular location or the entire country. How does it compare to datasets from other countries ?

Introduction is too long and some section should be removed or moved to methods. For instance paragraph 99-110 should be in methods.

Line 113, one of the largest city

Authors should explain how PCA analysis applied to the dataset helps advance the proposed method. If if other dimensionality reduction strategies can be applied.

Detailed descriptive statistics of the entire dataset should be provided.

All figures are not clear. More information should be provided on figures1-3 and explanation.

There are several types of plastics and it is not clear which one is the main identifier in MSW.

Reviewer #3: Main suggestion: The introduction needs to be revamped, as the paragraphs are not well connected and are difficult to follow for a first-time reader. The main observation in the results should be highlighted instead of presenting too many details. I also suggest assessing whether trend prediction using the data is possible.

Please consider refining the introduction and related works, as they are hard to follow; details and the main message are mixed, and some of the paragraphs are not well connected.

Line 61: mentions neural networks, but no such studies are presented in the paper. This raises expectations in the reader regarding the results.

Question: Has the author done any analysis with the 214 waste studies reported in line 99?

Lines 108-110: Please move this to the future section later, instead of making a promise upfront in the introduction.

Please highlight the seven parameters used in the studies (paper, glass,…) again in the Method section as it is lost in the introduction and makes it hard for the reader to understand the analysis.

Lines 236-239: Please clearly mention which figure; otherwise, it’s confusing for the reader to grasp the insight.

Suggestion to include descriptive captions in Figures 5 through 9.

The results section is cluttered with too many details. Please consider highlighting the main observation in each section; otherwise, the main insight will be lost in the details. Some of the analysis that does not carry important information could be moved to the supplementary section.

Question: What is the motivation behind correlating Manhattan and Euclidean distances in Figure 5? What necessitates studying the correlation between two metrics, as both are distance metrics explaining data in different ways, unless there is something important in the data that one metric cannot capture or is counter-intuitive?

After reading the paper, the title does not seem to be appropriate as it raises expectations when “machine learning techniques” is mentioned. The results are analyzed with regular statistical tools. Machine learning usually raises the expectation of predictive tools being used. One way to further improve the analysis is to also study the prediction analysis using the dataset, i.e., how well a model trained on these data predicts future trends. This could be tested with a recent sample test.

Suggestion for Improving the Abstract: The abstract should be improved as it conveys the main message of the paper.

Line 14: “found” >> “reveal/showed”

Line 15: “Comparison to disposal …” give more context on the data trends upfront.

Line 17: I would not call this a novel approach as such statistical tools are common and highly used in literature.

Line 19: Give a brief note on what conclusion was made.

Line 20: Give some context of time and distance here.

There are several grammatical/structural errors; please correct and refine those. Some noticeable suggestions:

Line 27: “many and increasing” >> “many increasing”

Line 31: “However” is not necessary.

Line 34: “The US is an exemplar…” >> “An example of such a case is the US.”

Line 45: mention briefly about the paper.

Lines 76-79: The message is not clear. Please simplify the structure.

Lines 80-81: Not clear.

Line 90: “; not….., however” >> “however, not…”

Lines 199-201: Not clear, consider rephrasing.

Lines 277-278: Not clear.

Lines 442-444: Rephrase the sentence, as “truer” does not sound proper in the context.

6. PLOS authors have the option to publish the peer review history of their article (what does this mean?). If published, this will include your full peer review and any attached files.

Reviewer #1: **Yes: **Dr. Ashootosh Mandpe

Reviewer #2: **Yes: **Jude A.Okolie

Reviewer #3: No

---

## [Author Response · Author response to Decision Letter 0]

10 Jul 2024

Response to comments

Editor 

Acknowledgements Do not include funding statement 

Corrected

Data Availability Minimal data provided 

Yes. All data are posted at: 

https://wastedata.info/data-repository; see the acknowledgement section.

Reviewer 1 

1 Global Use common (modern) English: clear, concise, academic tone 

Some language choices were made to emphasize approximate nature of waste composition studies; we think too many waste composition studies are overly precise. Thus our initial submission used more generalized terms (a quarter, or “approximately 15%” or “nearly 30%”) because we believe these show there are limits to how exact the measurements from these studies should be reported.

The comment was taken to heart, however, and numerous edits were made to tighten the language usage and use much more precise descriptions of the data (24.4% not “a quarter,” for instance). We decided this was not the battleground to make our stand regarding precision in waste characterizations. We also added more analytical elements to the paper.

2 Global Uniform font size and style (upper case, lower case) 

We revamped tables so that all material is now presented in Times-Roman 12 pt font. Not certain about the concern for upper case-lower case but we tried to be consistent throughout.

3 Title Rewrite title: clear & informative: Suggestion: determining spatiotemporal differences in waste generation practices using machine learning techniques 

Used: “Similarities and differences in waste composition over time and space determined by multivariate distance analyses”

4 Abstract More elaborative: explain how study was done w/o methodological details; summarize most important results & significance The abstract was rewritten to specifically address these points

5 Abstract No greater than 300 words 

Original abstract was 130 words; current abstract is 253 words.

6 Keywords More suitable: not multivariate, not disposal 

Modified: multivariate analyses; disposal trends

7 Keywords Replace waste studies with waste composition studies 

Done

8 Keywords Eliminate formatical mistake of using lower/upper case in writing key words 

Proper nouns were capitalized; common noun were not

9 Introduction Improve for clarity and conciseness 

Efforts were made to achieve this

10 26-27 Eliminate parentheses 

Parentheses were searched for generally across the paper and mostly removed

11 27 Eliminate “many & increasing” 

Edited to “causes environmental problems.”

12 28 Rewrite “greatest current concern” 

Done by combining lines 27 and 28

13 Introduction Provide examples from cited papers 

Added: “The value of collected materials for informal recycling is constrained by the metals and plastics content of the wastes [8]; the percentage and energy value of combustible components of the waste stream identifies systems where waste-to-energy incineration is a better option [9]; the organic, readily degradable content of landfilled MSW controls the generation of landfill gas, both as a potential pollutant and as an energy source [10]; and the composition of wastes affects the potential for development of more esoteric energy recovery processes, including pyrolysis, anaerobic digestion, and gasification [11].”

14 Introduction Eliminate parentheses (& shorten sentences) 

Parentheses were searched for generally across the paper and mostly removed.

15 99-146 Move to materials & methods 

Moved part of the material, eliminated much as repetitive.

16 Introduction Add brief statement of the overall aim of the work & whether that aim was achieved 

This was done: “Here we will examine four waste composition sort studies for a particular jurisdiction (New York City) conducted in 1990, 2004, 2013 and 2017. Because these studies of residential wastes not only sorted wastes collected for disposal but also recyclables it was possible to also depict the “generated” residential waste stream, which is not common in these kinds of studies. The percentage and rate data and associated trends for three broad materials categories (paper, food, and plastics) will be compared to EPA data and trends for the same years; the comparisons for disposal will be put into context using 88 other waste composition sort studies from 1988 to 2022. The sort study data trends generally show differences from the EPA data trends.

In addition, we will pioneer a more wholisitc waste stream composition approach that uses the basic techniques that drive machine learning. We will look at the multivariate distance relations among the result sets and use the principle that points close to each other in multivariate space are more similar and points that are farther away are different. This will allow us to underscore several common concepts about overall waste composition for different management processes (disposal and recycling) and to tentatively identify testable theses regarding changes in waste composition over time and space. These results will confirm that EPA modelled composition data tend to be different from composition results generated by physically sorting MSW.”

17 153 Justify exclusion of seasonal data 

Added: “The seasonal data sets were only available for half of the studies and generated data depiction complications, such as how to show annual values in the context of the four seasonal data points. The multiple closely packed data points across nine months made trend determination unduly complicated.”

18 172 Elaborate on 2 non-parametric exceptions 

Added: “…using PERMANOVA [Anderson 2005] to determine if significant differences existed across the entire waste stream. The first study determined that there were significant waste stream composition differences across three different waste districts in a Long Island municipality [34]. The second found that waste composition had not significantly changed for those districts two years later (in a broader examination of the impacts of changing from dual stream to single stream recycling) [35].”

19 616-617 Description of EPA data are ambiguous 

Edited sentences to: As with the NYC percentage data, EPA percentage data increased from 1990 to 2017. Unlike the NYC rate data, EPA rate data also showed increases in food waste generation and disposal.

20 625 Specify report cited here 

Added another citation to Thyberg et al (ref. 16)

21 Conclusion Needs to align with aims described at end of the revised Introduction (see comment 16) 

Original text addressed the first paragraph comment 16. This added material addressed the second paragraph of comment 16:

“It is clear that combining multivariate and standard analyses may improve our understanding of waste composition. The multivariate analyses can show broad patterns of similarity and differences for wastes as a whole based on the principle that points close to each other are similar and points farther way are different. Standard parameter comparisons can help illuminate exactly why the results are different or similar. Thus, disposal processes generate wastes that are different in overall character from recyclables separation. The way the overall character of disposed waste changes over time and distance is different from how recyclables change in character of time and distance. The multivariate analyses confirm that EPA modelled composition data are different in kind from composition results generated by physically sorting MSW.”

22 749 Too colloquial “mind you”; maintain academic tone 

True; that phrase deleted. 

23 Conclusion End with small paragraph regarding application of study & its future scope: how benefits real world in near future 

Done. Added: “Introduction of more wholistic waste stream assessments has the potential for immediate benefits in waste data studies and public planning. Clear identification of similar and different waste composition across jurisdictions adds to understanding of how programmatic factors such as different approaches to collection (Pay-as-You-Throw, single stream recycling) affect the basic units of waste. Being able to quantify the similarities and differences means that more sophisticated determinations of differences across waste systems can be considered. In New York State, where we have a project to assess the waste stream, state planners have created a tool to estimate waste composition based on simple land use classifications (rural, suburban, urban). We can use the wholistic assessments of waste composition to determine the degree that land use helps define waste composition and how other factors may be important, such as basic programmatic differences, or demography, or different approaches to public education. This may help focus and prioritize state efforts to achieve a more circular economy.

Reviewer 2 

1 Abstract Too generic: need study novelty, motivation, quantitative information 

The abstract was extensively rewritten

2 Introduction Interconnection between MSW composition and treatment/disposal methods and countries’ legislation 

Added: “Generally speaking, developing countries where organic materials especially food wastes are larger proportions of the waste stream tend to rely on landfilling and have less developed formal recycling programs; more developed countries where there may be more fabricated materials in the waste stream tend to have more organized recycling programs and may use incineration with energy recovery as a means of managing wastes. Often governments will specify the management of particular waste streams (such as yard wastes, food wastes, or recyclables) [3].” Ref 3 is What a waste II which contains regional waste management program discussions that support the assertions.

3 Introduction Explain scope of EPA dataset better 

Added: EPA explicitly states that the model is meant to represent the US waste stream as a whole and is unlikely to be representative of MSW at any particular location (EPA 2014). Ref is to EPA methodology report.

4 99-110 Move to methods 

Eliminated as repetitious, moved following paragraphs into materials and methods and edited the resulting section.

5 133 Correct to “one of the largest” 

No – it is the largest (2x #2, LA)

6 Introduction Explain how PCA helps the analysis – any other dimensional reduction strategies can be applied? 

This was in the original introduction and was moved to the Materials and Methods section. We think it explains that PCA use was a choice we made and that other dimensional reduction techniques could have been used. We like PCA.

“In addition, statistical analyses such as multidimensional scaling (MDS), principal component analysis (PCA), and various forms of cluster analysis lend themselves to simplified visual depictions of similarities and differences across multivariate data sets such as the waste composition studies. Reducing the dimensionality of multi-dimensional data sets each have relative advantages and issues. Multivariate correlational analyses can also find similarities and differences across these kinds of data sets [33]. Here we use the distance relations and principal component analysis (PCA) to find broader similarities and differences among the four NYC studies and four EPA composition data sets from the same time periods.”

7 General Detailed descriptive statistics of the entire data set should be provided 

Added more descriptive statistics in the Tables and added Tables 5-7. We think this is useful and adds to our work. The original data from the NYC data do and do not have attached descriptive statistics in them (the 1990 report does not but there are pages and pages of data for 2004, 2013, 2017. EPA data do not have descriptive statistics (model outputs). Descriptive statistics are not possible where we combined NYC data sets (because we do not have access to the sample data and the exact means by which the sample data were translated to borough and City-wide data sets. We think simply reproducing the NYC stats is not useful for our analyses. Expanding this effort to cover all of the comparative studies would be a huge effort. We think we have addressed some of the comment but think we do not understand the scope of the effort that this comment is trying to achieve. 

8 General All figures not clear. Improve descriptions of Fig 1-3 

We met the journal figure requirements; the figures as reproduced in the PDF do not reflect the actual figure quality. Please see the .tiff files. We think the figures are clear as we size them (the labels on the heat maps are small but clear). We like our captions for Fig (new numbering) 1, 3 and 4. 

9 General Which plastic is referred to 

Plastic here is not a specific plastic but refers to all plastics (containers, films, durable products); see the appendix for materials groups inclusions and exclusions. Added this in Materials and Methods: “The categories include all subcategories (if any) of the waste sorts. So “plastics” here includes all containers of various resin types, plastic films, and all durable plastics.”

Reviewer 3 

Introduction Revamp: paragraphs not well connected 

The introduction was extensively rewritten.

Results Highlight main observation (too many details) 

Worked hard to make the focus better

General Trend prediction possible? 

Not our intention to make predictions

61 Mentions neural networks but do not include such results 

We wanted to acknowledge that some are using this technique; we are not (yet) impressed with the results.

99 Analyses of the 214 studies done? 

No, and deleted this reference.

108-110 Promise of future release not appropriate here 

Deleted this section

Methods Highlight the seven parameters again (lost from introduction) 

Done

236-239 Overall waste generation rates: which figure? 

They are in ref 38. Added relevant figures in File S6

Figures 5-9 Improve captions (more descriptive) 

We think the captions are adequate. 

Results Too cluttered with details 

Agreed; we simplified the discussions.

Results Why the section on the correlation between Manhattan and Euclidean distances 

Agreed. It was removed to the Supplemental files S5.

Title Not appropriate; results analyzed by std. stats. Machine learning raises expectation of predictive tools; is this intended to be predictive (if so, provide an example) 

We think of machine learning as a grouping process: it is used to determine whether things are similar or different, at the very basic level. The algorithms for doing this are often hidden due to the process used to determine these similarities. For applications like Chat GPT, the models generate members of a class that can be anticipated to follow on a particular prompt; we don’t think this “predictive.” Here we would use the grouping process to determine if a set of waste composition data was similar to (close to) or different (far from) another result or set of results. We think of this as a classification process not predictions. 

But the point is taken. We have removed (nearly) all references to machine learning

14 Found should be revealed/showed 

Substituted showed

15 More context on data trends upfront (comp to 88 other studies) 

Abstract was revised to include more details

17 Not novel; commonly used 

Disagree; multivariate distance measures commonly used in many fields but never before in waste composition studies

19 Specify conclusions that were reached 

Abstract was revised to include more details.

20 Context of “time & distance” 

Specified that they applied to the NYC data sets

27 Many & increasing: many increasing 

Deleted the clause

31 However: eliminate 

Done

34: US is an exemplar: An example of such a case is the US 

Removed the sentence.

45 Expand about the cited paper 

This is covered in the discussion; would be repetitive to do it here too.

76-79 Explain derivation of rates better 

Added: “That is, given a percentage value for a particular material, the amount of that material can be determined by multiplying the percentage by the total amount of wastes. Dividing the resultant amount of wastes by the relevant population and a time measure, such as 365 for a daily value, will create a rate datum.”

80-81 Not clear (extraction of data across studies) 

Edited to: “Thus, it is possible to collect a se

---

## [Editor Report · Decision Letter 1]

23 Jul 2024

Similarities and differences in waste composition over time and space determined by multivariate distance analyses

PONE-D-24-14194R1

Dear Dr. Tonjes,

We’re pleased to inform you that your manuscript has been judged scientifically suitable for publication and will be formally accepted for publication once it meets all outstanding technical requirements.

Kind regards,

Basant Giri, Ph.D.

Academic Editor

PLOS ONE
---

## [Editor Report · Acceptance letter]

24 Sep 2024

PONE-D-24-14194R1 

PLOS ONE

Dear Dr. Tonjes, 

I'm pleased to inform you that your manuscript has been deemed suitable for publication in PLOS ONE. Congratulations! Your manuscript is now being handed over to our production team.

Kind regards, 

on behalf of

Dr. Basant Giri 

Academic Editor

PLOS ONE